# Causally Evaluating the Learnability of Formal Language Tasks

**Vésteinn Snæbjarnarson** [1 2] **Anej Svete** [1] **Josef Valvoda** [* 2] **Reda Boumasmoud** [1] **Brian DuSell** [1] **Ryan Cotterell** [1]

## Abstract

Language models, as multi-task learners, acquire a wide range of abilities during training. A fundamental question is how much task-specific data is needed to learn a given task. Answering this for natural language is difficult: tasks are hard to delineate and can confound one another. To rigorously investigate the relationship between data frequency and learnability, we turn to a controlled setting using formal languages induced from probabilistic finite automata. These serve as a methodological testbed to demonstrate that standard correlational evaluation practices are inherently flawed. To enable causal analysis, we introduce the *binning semiring*, an algebraic object that lets us control how often a targeted property occurs in a sampled corpus. We formulate the experimental pipeline as a causal graphical model and derive decomposed Kullback–Leibler divergence metrics to measure the learnability of specific sub-tasks. Our experiments show that evaluating learnability without causal intervention leads to incorrect conclusions due to confounders in correlational analysis, and serve as a warning about correlational pitfalls in natural-language settings.

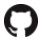 https://github.com/vesteinn/causal-eval-formal-languages

## 1. Introduction

Training a language model (LM) is, in effect, a massive multi-task learning problem. From a single corpus, a modern LM learns to translate languages (e.g., English into Dutch; Du et al., 2025), generate Python code (Chen et al., 2021; Li et al., 2023; Rozière et al., 2024), write job applications (Wiles et al., 2025), assist with mathematical proofs (Lewkowycz et al., 2022) *inter multa alia*. A natural question is then: how much task-specific data does an LM need to learn a given task? The answer surely depends on how similar the task is to others in the training data—a programming language with Python-like syntax may be acquired from a few examples given the prevalence of Python in the training data. In contrast, a programming language with an unfamiliar paradigm may demand substantially more. However, determining precisely how much training data a particular task requires in a multi-task setting is difficult: in a large natural-language corpus, tasks are hard to delineate and label, and different tasks can share underlying similarities, introducing confounders. We must isolate and control these confounders to determine how data scarcity affects task performance.

The goal of this paper is to replicate a multi-task learning problem in a controlled setting. To do so, we turn to formal languages, a common testbed for studying the learnability of neural architectures (Lake and Baroni, 2018; Michalenko et al., 2019; Hupkes et al., 2020; Ruis et al., 2020; Valvoda et al., 2022; Delétang et al., 2023; Svete et al., 2024a; Van der Poel et al., 2025; Butoi et al., 2025). Specifically, we study how well we can train an LM to approximate a probabilistic finite automaton (PFA). When learning a PFA, each symbol, state, and transition of the underlying automaton can be seen as a task, and these tasks are interrelated through the automaton's structure. Inter-task confounders, therefore, arise in the formal-language learning setting as well. Yet, the standard methodology for evaluating language models' ability to learn probabilistic formal languages does not account for them (Borenstein et al., 2024; Someya et al., 2025).

As an illustrative example, consider the PFA in Fig. 1. It assigns non-zero probability to two kinds of strings: those of the form aa*, and those containing an odd number of b's—a probabilistic version of PARITY (Hahn and Rofin, 2024) prefixed by b. Theory suggests that transformers easily learn aa* (Yang et al., 2024; Li and Cotterell, 2026) but not PARITY (Hahn and Rofin, 2024). The parameter $\eta$ governs which branch the PFA enters, and thus how often each task appears in a corpus sampled from it. But $\eta$ does more than control task frequency: it also shifts the overall distribution the model must learn, including the expected string length, since the two branches yield strings of different expected lengths. So when a model trained on a corpus with few PARITY strings performs poorly on

---

*Now at Google. [1]ETH Zürich [2]University of Copenhagen. Correspondence to: Vésteinn Snæbjarnarson <vest.snae@gmail.com>, Ryan Cotterell <ryan.cotterell@inf.ethz.ch>.

*Proceedings of the 43rd International Conference on Machine Learning*, Seoul, South Korea. PMLR 306, 2026. Copyright 2026 by the author(s).

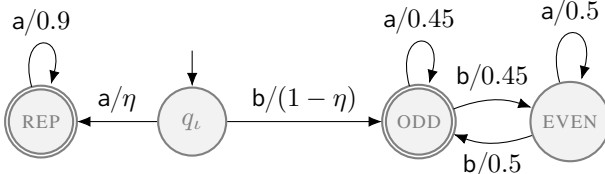

*Figure 1.* A PFA recognizing the language aa* ∪ ba*(ba*ba*)*. The transitions left of $q_\iota$ correspond to the star-free language aa* (which is easier for transformers to learn), and the ones to the right correspond to the PARITY language (which is harder for transformers to learn). If $\eta \in [0, 1]$ is low, we might wrongly assume that aa* is harder to learn.

PARITY, we cannot tell whether the difficulty stems from there being too few PARITY strings to learn from, the shift in the corpus's length distribution that co-occurs with the change in $\eta$, or the fact that PARITY may be inherently hard for this architecture. Disentangling these requires causal intervention (Pearl et al., 2016).

To carry out this disentanglement, we contrast two ways of generating data from a given weighting of a PFA, distinguishing the *causal* effect of certain string properties on learnability—e.g., how often a string causes its PFA to pass through a given transition—from the *correlational* one. The *correlational* setup is the one implicit in standard practice: sample many weightings of the PFA, draw a corpus from each, train a language model on each corpus, and correlate the realized count of the target property with the trained model's performance. The *causal* setup, by contrast, intervenes directly on the sampling procedure to produce a corpus with an exact count for a chosen transition under any given weighting of the automaton, decoupling the count from the rest of the corpus's properties. In both setups, we measure task-specific performance with a bespoke decomposition of the Kullback–Leibler divergence (App. G) between the trained model and the PFA and plot it against the occurrence count. We show results for a transformer (Vaswani et al., 2017) and an LSTM (Hochreiter and Schmidhuber, 1997) in Fig. 2. Even in this simple setting, the causal (solid) and correlational (dotted) curves differ. The setup also recovers the known LSTM–transformer gap on PARITY (Hahn and Rofin, 2024): in the ODD panel of Fig. 2, the LSTM's per-state Kullback–Leibler divergence drops about an order of magnitude below the transformer's as the PARITY count grows, whereas the transformer's curve plateaus. Unlike prior work that only argues for the gap's existence, we trace it directly as a function of PARITY occurrences.

On the theoretical side, to enable causal evaluation of how well language models can learn PFAs, we introduce a new algebraic tool: the binning semiring (§4.1).[1] It lifts each transition weight into a vector indexed by the

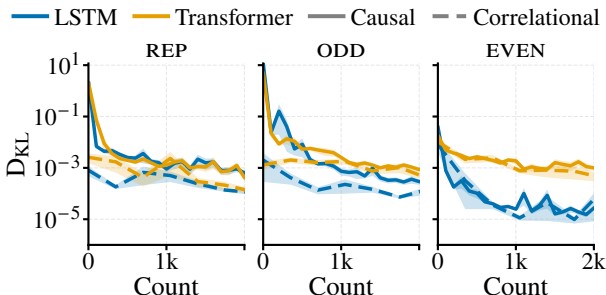

*Figure 2.* Each panel plots state-wise decomposed Kullback–Leibler divergence ($D_{KL}$; lower is better) against the number of times the target state occurs in training. *Correlational* (dotted): sample many weightings of the PFA in Fig. 1 and record the realized count. *Causal* (solid): intervene in the sampling procedure to sample specific counts. Gaps reflect confounding from the joint distribution over weights.

number of target occurrences along a path, so summing path weights gives the conditional distribution over corpora with a given count—exactly the distribution the causal setup samples from. We present methods (§4) for efficiently sampling datasets from a PFA such that a target property is represented exactly $N$ times among $K$ samples.

Empirically, we conduct experiments in three settings to show that causal evaluation matters when studying how well language models learn PFAs. (i) We vary weights over the PFA of Fig. 1 (already shown in Fig. 2), which combines probabilistic versions of a star-free language and PARITY. (ii) We sample a large set of PFAs with dozens of states to consider the learnability of properties independent of the specific configuration of the PFA encoding them. (iii) We consider fixed PFA topologies while varying the weightings. Across all three settings, correlational estimates of task learnability systematically diverge from their causal counterparts; in setting (ii), the correlational trend even *inverts* (Fig. 5), indicating that automata that naturally produce few instances of a task tend to have structural properties that confound learnability estimates. These results demonstrate that standard correlational methodology can spoil conclusions about formal language learnability.

## 2. Preliminaries

**Basics.** We write $\mathbb{N} \stackrel{\text{def}}{=} \{1, 2, 3, \ldots\}$ for the natural numbers. For $N \in \mathbb{N}$, we write $[N] \stackrel{\text{def}}{=} \{1, \ldots, N\}$ and $[N]_0 \stackrel{\text{def}}{=} \{0, 1, \ldots, N\}$.

**Language Models.** An **alphabet** $\Sigma$ is a finite non-empty set of **symbols**. The **Kleene closure** $\Sigma^*$ is the set of all strings of symbols from $\Sigma$. A **language model** (LM) $p$ is a probability distribution over $\Sigma^*$. The **prefix probability**[2]

---

[1]While the binning semiring can be used on any model whose computations factor over a semiring, our study focuses on PFAs.

[2]Note that prefix probabilities do *not* define a distribution over $\Sigma^*$, as they do not sum to 1.

$\overrightarrow{p}(\boldsymbol{\sigma})$ of a string $\boldsymbol{\sigma} \in \Sigma^*$ is the total probability of strings starting with $\boldsymbol{\sigma}$ under $p$; the **conditional** prefix probability $\overrightarrow{p}(\boldsymbol{\sigma}' \mid \boldsymbol{\sigma})$ extends this to a continuation $\boldsymbol{\sigma}' \in \Sigma^*$:

$$\overrightarrow{p}(\boldsymbol{\sigma}) \stackrel{\text{def}}{=} \sum_{\boldsymbol{\sigma}' \in \Sigma^*} p(\boldsymbol{\sigma}\boldsymbol{\sigma}'), \quad \overrightarrow{p}(\boldsymbol{\sigma}' \mid \boldsymbol{\sigma}) \stackrel{\text{def}}{=} \frac{\overrightarrow{p}(\boldsymbol{\sigma}\boldsymbol{\sigma}')}{\overrightarrow{p}(\boldsymbol{\sigma})}, \quad (1)$$

with $\overrightarrow{p}(\boldsymbol{\sigma}' \mid \boldsymbol{\sigma}) \stackrel{\text{def}}{=} 0$ when $\overrightarrow{p}(\boldsymbol{\sigma}) = 0$. Many LMs define $p(\boldsymbol{\sigma})$ autoregressively via

$$p(\boldsymbol{\sigma}) \stackrel{\text{def}}{=} \overrightarrow{p}(\text{EOS} \mid \boldsymbol{\sigma}) \prod_{t=1}^{|\boldsymbol{\sigma}|} \overrightarrow{p}(\sigma_t \mid \boldsymbol{\sigma}_{<t}), \quad (2)$$

where $\text{EOS} \notin \Sigma$ is a designated end-of-string symbol. We write $\Sigma_{\text{EOS}} \stackrel{\text{def}}{=} \Sigma \cup \{\text{EOS}\}$ for the extended alphabet. We denote a neural LM as $p_{\boldsymbol{\theta}}$, where $\boldsymbol{\theta}$ collects the neural LM's learnable parameters.

**Algebra and Formal Languages.** Let $\mathbb{K}$ be a set, $\odot$ a binary operation, and $\mathbf{1} \in \mathbb{K}$. We say that $(\mathbb{K}, \odot, \mathbf{1})$ is a **monoid** if *(i)* $\mathbb{K}$ is closed under $\odot$; *(ii)* $\odot$ is associative; and *(iii)* $\mathbf{1}$ is the unit of $\odot$: $\forall a \in \mathbb{K} : \mathbf{1} \odot a = a \odot \mathbf{1} = a$. We say that a monoid is **commutative** if $\forall a, b \in \mathbb{K} : a \odot b = b \odot a$. A **semiring** is a tuple $(\mathbb{K}, \oplus, \otimes, \mathbf{0}, \mathbf{1})$ where *(i)* $(\mathbb{K}, \oplus, \mathbf{0})$ is a commutative monoid; *(ii)* $(\mathbb{K}, \otimes, \mathbf{1})$ is a monoid; *(iii)* $\otimes$ left- and right-distributes over $\oplus$: $a \otimes (b \oplus c) = (a \otimes b) \oplus (a \otimes c)$ and $(b \oplus c) \otimes a = (b \otimes a) \oplus (c \otimes a)$; and *(iv)* $\otimes$ with $\mathbf{0}$ annihilates $\mathbb{K}$: $\mathbf{0} \otimes a = a \otimes \mathbf{0} = \mathbf{0}$. The operation $\otimes$ is referred to as **multiplication** and $\oplus$ as **addition**. A semiring admitting well-behaved infinite sums is called **complete**, i.e., in a complete semiring, we can define **asteration** $a^* \stackrel{\text{def}}{=} \bigoplus_{i=0}^{\infty} a^i = \bigoplus_{i=0}^{\infty} \bigotimes_{j=1}^{i} a$. The **real semiring** is $(\mathbb{R}, +, \times, 0, 1)$ with the standard addition and multiplication. Adjoining $\infty$ to positive reals in the natural way gives a complete semiring with $a^* = \frac{1}{1-a}$ for $a < 1$ and $a^* = \infty$ otherwise.

**Finite Automata.** A **weighted finite automaton** (WFA) $\mathcal{A}$ over a semiring $(\mathbb{K}, \oplus, \otimes, \mathbf{0}, \mathbf{1})$ is a tuple $(Q, \Sigma, \delta, \lambda, \rho)$ where $Q$ is a finite set of states; $\Sigma$ is an alphabet; $\delta \colon Q \times \Sigma \times Q \to \mathbb{K}$ is a transition function, where we say that $\mathcal{A}$ has transition $p \xrightarrow{a/w} q$ with weight $w$ if $\delta(p, a, q) = w$; and $\lambda \colon Q \to \mathbb{K}$ and $\rho \colon Q \to \mathbb{K}$ are the initial and final weight functions, respectively. A **path** $\boldsymbol{\pi}$ in $\mathcal{A}$ is a finite sequence of transitions connected by states: $q_0 \xrightarrow{a_1/w_1} q_1 \cdots q_{m-1} \xrightarrow{a_m/w_m} q_m$. We write $|\boldsymbol{\pi}| = m$ for $\boldsymbol{\pi}$'s length. The **inner weight** of $\boldsymbol{\pi}$ is $\overline{w}(\boldsymbol{\pi}) \stackrel{\text{def}}{=} w_1 \otimes \cdots \otimes w_m$, its **weight** is $\boldsymbol{w}(\boldsymbol{\pi}) \stackrel{\text{def}}{=} \lambda(q_0) \otimes \overline{w}(\boldsymbol{\pi}) \otimes \rho(q_m)$, and its **yield** is $a_1 \cdots a_m$. $\Pi(\mathcal{A})$ denotes the set of all paths in $\mathcal{A}$. We say that $\mathcal{A} = (Q, \Sigma, \delta, \lambda, \rho)$ is **deterministic** (a DPFA) if, for every $p \in Q, \sigma \in \Sigma$, there is at most one $q \in Q$ such that $p \xrightarrow{\sigma/w} q \in \delta$ with $w \neq \mathbf{0}$, and there is a single state $q_\iota$ with $\lambda(q_\iota) \neq \mathbf{0}$. In this case, we refer to $q_\iota$ as the **initial**

**state**. A deterministic WFA can have at most one path with non-$\mathbf{0}$ path weight yielding a string $\boldsymbol{\sigma} \in \Sigma^*$ from the initial state $q_\iota$. The **backward weight** $\beta_{\mathcal{A}}(q)$ of a state $q \in Q$ is the sum of the weights of all paths starting at $q$:

$$\beta_{\mathcal{A}}(q) \stackrel{\text{def}}{=} \bigoplus_{\boldsymbol{\pi} \in \Pi(\mathcal{A}), \iota(\boldsymbol{\pi}) = q} \overline{w}(\boldsymbol{\pi}) \otimes \rho(\varphi(\boldsymbol{\pi})), \quad (3)$$

where $\iota(\boldsymbol{\pi})$ and $\varphi(\boldsymbol{\pi})$ denote the first and last states of $\boldsymbol{\pi}$, respectively. The **allsum** of a DPFA $\mathcal{A}$ is $Z \stackrel{\text{def}}{=} \sum_{\boldsymbol{\pi} \in \Pi(\mathcal{A})} \boldsymbol{w}(\boldsymbol{\pi}) = \beta_{\mathcal{A}}(q_\iota)$.

**Probabilistic Automata.** We say that a real-weighted WFA with non-negative weights is **probabilistic** (a PFA) if $\sum_{q \in Q} \lambda(q) = 1$ and, for all states $q \in Q$, $\rho(q) + \sum_{q \xrightarrow{\sigma/w} q' \in \delta} w = 1$. A PFA induces a probability distribution over $\Sigma^*$, where the probability of a string $\boldsymbol{\sigma}$ is the sum of the weights of all paths yielding $\boldsymbol{\sigma}$ from the initial state $q_\iota$. For a DPFA, each string $\boldsymbol{\sigma} \in \Sigma^*$ admits at most one path; let $\widehat{\delta}(\boldsymbol{\sigma}) \in Q$ denote the state reached by reading $\boldsymbol{\sigma}$ from $q_\iota$. A DPFA then realizes an autoregressive language model (Eq. (2)) whose next-symbol distribution at history $\boldsymbol{\sigma}$ is supported on $\Sigma_{\text{EOS}}$: $\overrightarrow{p}(\sigma \mid \boldsymbol{\sigma}) = w$ when $\widehat{\delta}(\boldsymbol{\sigma}) \xrightarrow{\sigma/w} q' \in \delta$ and $\overrightarrow{p}(\text{EOS} \mid \boldsymbol{\sigma}) = \rho(\widehat{\delta}(\boldsymbol{\sigma}))$.

**Graphical Causal Models.** We now introduce graphical causal models and interventions. We adopt the following notation: sets are denoted with uppercase calligraphic font, e.g., $\mathcal{X}$ and $\mathcal{Y}$; subsets of a set $\mathcal{X}$ are uppercase letters, e.g., $X$; and elements of a set $\mathcal{X}$ are lowercase letters, e.g., $x$. Random variables over $\mathcal{X}$ are typeset as uppercase, unitalicized letters, e.g., X. Bold uppercase letters, e.g., $\boldsymbol{X}$, are used to denote sets of random variables. A **graphical causal model** (GCM) consists of a directed acyclic graph (DAG) $\mathcal{G} = (\boldsymbol{V}, \boldsymbol{E})$ whose nodes $\boldsymbol{V} = \{V_1, \ldots, V_N\}$ are random variables over domains $\mathcal{V}_1, \ldots, \mathcal{V}_N$ and whose edges $\boldsymbol{E} \subseteq \boldsymbol{V} \times \boldsymbol{V}$ encode direct causal influence. Each node carries a conditional distribution $\mathbb{P}(V_n \mid \text{PA}_n)$ given its **parent variables** $\text{PA}_n \subseteq \boldsymbol{V}$ in $\mathcal{G}$. The joint distribution then factorizes over the DAG as

$$\mathbb{P}(V_1, \ldots, V_N) = \prod_{n=1}^{N} \mathbb{P}(V_n \mid \text{PA}_n). \quad (6)$$

The graphical model allows one to reason about **interventions**—changes to the joint distribution that affect individual variables in the model by modifying their direct causes. Operationally, intervening on X replaces its conditional distribution $\mathbb{P}(X \mid \text{PA}_X)$ with a point mass at the intervened value, leaving all other conditionals untouched. Standard notation for such operations is $\mathbb{P}(Y \mid \text{do}(X = X))$, which denotes the distribution of Y when X is set to event $X$ (Pearl et al., 2016). For this to be well-defined, we ensure that the

$$\mathbb{P}(M \mid D \in \mathcal{P}) = \int_{\mathcal{A}} \int_{\mathcal{L}} \int_{\mathcal{P}} \mathbb{P}(M \mid da, dl)\mathbb{P}(dl \mid dd)\mathbb{P}(dd \mid da, D \in \mathcal{P})\mathbb{P}(da \mid D \in \mathcal{P}) \quad (4)$$

$$\mathbb{P}(M \mid \mathrm{do}(D \in \mathcal{P})) = \int_{\mathcal{A}} \int_{\mathcal{L}} \int_{\mathcal{P}} \mathbb{P}(M \mid da, dl)\mathbb{P}(dl \mid dd)\mathbb{P}(dd \mid da)\mathbb{P}(da) \quad (5)$$

*Figure 3.* **Left:** Graphical causal model for evaluating the effect of intervening on the *dataset* D on model *performance* M, given a PFA language model A and *trained model* L. **Right:** Conditioning (Eq. (4)) induces $\mathbb{P}(a \mid D \in \mathcal{P})$; intervening (Eq. (5)) restores the marginal $\mathbb{P}(a)$. The variables are described in §3.

event $X$ has a positive probability. [3]

Importantly, the GCM formulation above allows us to handle intervention distributions in continuous spaces through covariate adjustment formulas as described in Peters et al. (2017, Section 6.6). Specifically, if Z denotes the parents of X, we can solve for the causal effect of setting X to $X$ using the adjustment integral:

$$\mathbb{P}(Y \mid \mathrm{do}(X = X)) = \int \mathbb{P}(Y \mid X = X, Z)\mathbb{P}(dz). \quad (7)$$

Intuitively, we average over the natural distribution of X's parents while substituting the intervened value $X$ in the conditional. Even if we cannot directly compute Eq. (7), we can approximate it via sampling. In the next section, we apply this framework to model interventions on sampled datasets.

## 3. A Causal Model for Learnability

We now develop a causal framework to isolate the effects of dataset properties on language-model performance. We instantiate the graphical causal model framework with a specific DAG, shown on the left in Fig. 3, defined over four random variables $V = \{A, D, L, M\}$. These variables correspond to our experimental pipeline and are described below.

**PFA:** A is a random variable over the space of PFA language models $\mathcal{A}$, representing the generating process.

**Dataset:** D is a random variable over the domain of datasets $\mathcal{D}$, generated by the PFA.

**Learned Model:** L is a random variable over the parameter space $\mathcal{L}$ (e.g., neural weights), resulting from training.

**Performance:** M is a random variable in $\mathcal{m} \subseteq \mathbb{R}$ that represents the fidelity of the learned model.

---

[3]However, we can easily generalize to the case where $X$ has probability zero by defining each conditional $\mathbb{P}(V_n \mid \mathrm{PA}_n)$ as a **Markov kernel**, a measure-theoretic surrogate for conditional probability that avoids conditioning on measure-zero events; see App. C. In the body of the paper we treat these conditionals as conditional densities, which is only rigorous whenever the joint distribution is absolutely continuous with respect to a product reference measure (Lebesgue on continuous coordinates, counting on discrete ones).

The causal structure $\boldsymbol{E}$ is defined by the dependencies between these stages: the dataset depends on the PFA (A $\rightarrow$ D), the learned model depends on the training data (D $\rightarrow$ L), and the performance measure depends on both the learned model and the PFA ({L, A} $\rightarrow$ M). This implies the following factorization of the joint distribution:

$$\mathbb{P}(A, D, L, M) = \mathbb{P}(A)\mathbb{P}(D \mid A)\mathbb{P}(L \mid D)\mathbb{P}(M \mid L, A). \quad (8)$$

**Intervening vs. Conditioning.** We define a *property* $\mathcal{P} \subseteq \mathcal{D}$ as a set of datasets. In this paper, given a PFA $\mathcal{A}$, we specify properties in terms of how often a designated subset of the automaton's transitions is traversed, e.g., "the dataset was sampled by traversing a specific set of transitions exactly $N$ times." This allows us to specify properties such as "the dataset contains exactly $N$ occurrences of symbol $\sigma$" as a special case. Specifically, given a property $\mathcal{P}$, our goal is to distinguish the following cases with our causal model.

(1) **Conditioning.** Evaluate $\mathbb{P}(M \mid D \in \mathcal{P})$, which introduces a dependence between the PFA language model A and the dataset D (the term $\mathbb{P}(a \mid D \in \mathcal{P})$ in Fig. 3). PFAs that are more likely to produce datasets in $\mathcal{P}$ will be overrepresented.

(2) **Intervening.** Evaluate $\mathbb{P}(M \mid \mathrm{do}(D \in \mathcal{P}))$ by modifying the data generation process to satisfy $\mathcal{P}$, preserving the original distribution $\mathbb{P}(a)$.

To compare these two cases empirically, we (i) sample a set of PFAs; (ii) sample corpora from each, either conditionally on $D \in \mathcal{P}$ or under the intervention $\mathrm{do}(D \in \mathcal{P})$; (iii) train a language model on each corpus; and (iv) measure its performance against the PFA. The expected performance under each setup gives the two estimands:

$$\mu_{\mathrm{int}}(\mathcal{P}) \stackrel{\mathrm{def}}{=} \mathbb{E}[M \mid \mathrm{do}(D \in \mathcal{P})], \quad (9a)$$

$$\mu_{\mathrm{obs}}(\mathcal{P}) \stackrel{\mathrm{def}}{=} \mathbb{E}[M \mid D \in \mathcal{P}]. \quad (9b)$$

Thus, $\mu_{\mathrm{int}}(\mathcal{P}) - \mu_{\mathrm{obs}}(\mathcal{P})$ isolates the effect of enforcing $\mathcal{P}$ while preserving the original distribution of A (cf. Fig. 3); the two estimands are derived from the same end-to-end pipeline and differ only in how the corpus is sampled.

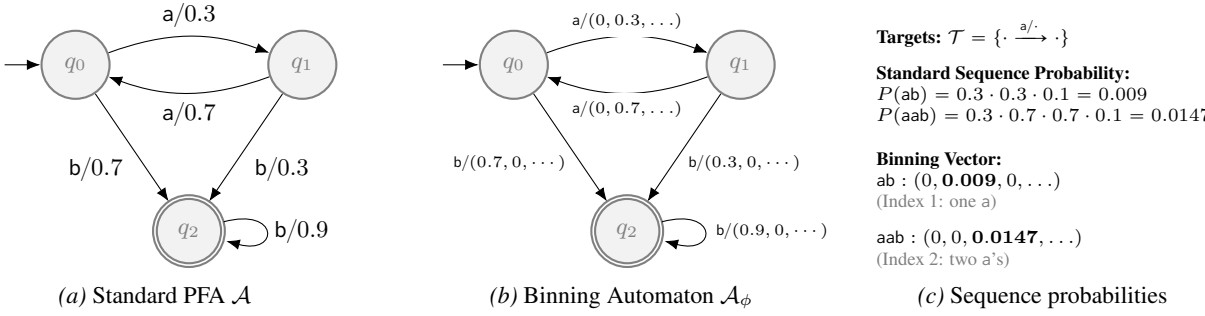

*(a)* Standard PFA $\mathcal{A}$          *(b)* Binning Automaton $\mathcal{A}_\phi$          *(c)* Sequence probabilities

**Figure 4.** Given an automaton $\mathcal{A}$ (a), we lift its weights to the *binning semiring*. The lifted *binning automaton* (b) has vectors $\boldsymbol{v} = (v_0, v_1, \ldots, v_R)$ as weights. Transitions on symbol a $\in \mathcal{T}$ shift probability mass to the next index, while b $\notin \mathcal{T}$ stays in place. (c) Multiplication of these vectors (elements of the binning semiring) is defined such that it corresponds to counting occurrences.

## 4. Sampling Under Event Constraints

To implement the causal intervention of §3, we need to sample from $\mathbb{P}(\mathrm{D} \mid \mathrm{A} = a, \mathrm{do}(\mathrm{D} \in \mathcal{P}))$ for a given automaton $a = \mathcal{A}$, i.e., draw a dataset D from $\mathcal{A}$ subject to a property $\mathcal{P}$. For example, we may wish to draw a dataset that contains symbol a exactly $N$ times. We first clarify what we mean by a property $\mathcal{P}$. Each sampled string from $\mathcal{A}$ is obtained by sampling an accepting path in the automaton, so we can relate each string structurally to the transitions it traverses. In the case of a DPFA, there is exactly one sequence of transitions for every string in the language. Given $N \in \mathbb{N}$ and a set of transitions $\mathcal{T}$, we consider two intervention types that we use throughout §5: the **exact-count** intervention fixes the corpus's total number of $\mathcal{T}$-transitions to $N$, and the **at-least-once** intervention fixes the number of strings (out of $K$) that contain at least one $\mathcal{T}$-transition. In both cases, we write $\mathcal{P} \subseteq \mathcal{D}$ for the set of datasets meeting the chosen constraint, and the intervention enforces it. [4] By controlling the occurrences of particular sets of transitions, we can target, e.g., all transitions that emit a specific symbol, or all transitions out of a given state.

### 4.1. The Binning Semiring

To construct an efficient sampling algorithm, we define the **binning semiring**, an algebra over a base semiring that tracks occurrence counts of a targeted feature through path computations. For a base semiring $(\mathbb{K}, \oplus, \otimes, \boldsymbol{0}, \boldsymbol{1})$ and threshold count $R \in \mathbb{N}$, the $R^{\text{th}}$-order binning semiring $[\![\mathbb{K}]\!]_R = (\mathbb{K}^{R+1}, \circledplus, \circledtimes, \circledcirc{0}, \circledcirc{1})$ has as its underlying set the $(R + 1)$-tuples

$$\boldsymbol{v} = (v_0, v_1, \ldots, v_R), \qquad v_n \in \mathbb{K}, \qquad (10)$$

where, for $n < R$, element $v_n$ records the weight associated with $n$ target occurrences, and $v_R$ aggregates all counts greater than or equal to $R$. Addition $\circledplus$ is element-wise,

with $\circledcirc{0} \stackrel{\text{def}}{=} (\boldsymbol{0}, \ldots, \boldsymbol{0})$. Multiplication $\circledtimes$ is an overflow-accumulating convolution: for $\boldsymbol{v}, \boldsymbol{u} \in \mathbb{K}^{R+1}$,

$$(\boldsymbol{v} \circledtimes \boldsymbol{u})_n \stackrel{\text{def}}{=} \begin{cases} \sum_{j=0}^n v_j \otimes u_{n-j} & \text{if } n < R, \\ \displaystyle\sum_{\substack{i+j \geqslant R, \\ 0 \leqslant i,j \leqslant R}} v_i \otimes u_j & \text{if } n = R, \end{cases} \quad (11)$$

with $\circledcirc{1} \stackrel{\text{def}}{=} (\boldsymbol{1}, \boldsymbol{0}, \ldots, \boldsymbol{0})$. Below $R$, $\circledtimes$ is the ordinary convolution that adds occurrence counts; at the top coordinate it absorbs all overflow, so any product with combined count $\geqslant R$ accumulates there.[5] For $\boldsymbol{v} \in \mathbb{K}^{R+1}$ and $n \in \mathbb{N}$, we write $\boldsymbol{v}^{\circledtimes n} \stackrel{\text{def}}{=} \underbrace{\boldsymbol{v} \circledtimes \cdots \circledtimes \boldsymbol{v}}_{n \text{ times}}$. Following convention, we sometimes write $[\![\mathbb{K}]\!]_R$ to mean $\mathbb{K}^{R+1}$.

**Time complexity.** Assuming $\oplus$ and $\otimes$ run in constant time, $\circledplus$ runs in time $\mathcal{O}(R)$. The $\circledtimes$ operation (a convolution) runs in time $\mathcal{O}(R^2)$. In the special case of real-weighted WFAs, however, we can implement $\circledtimes$ using the fast Fourier transform (FFT), in which case it runs in $\mathcal{O}(R \log R)$ time. Despite the faster implementation of $\circledtimes$ with FFT, we do not adopt it due to underflow for large values of $R$. Instead, we use the log semiring for numerical stability. In practice, we implement these operators using vectorized operations in PyTorch (Paszke et al., 2019), allowing us to run them efficiently on GPUs.

### 4.2. Binning Automaton

We now describe how to **lift** a PFA $\mathcal{A}$ to a WFA over the binning semiring as shown in Fig. 4. Let $\mathcal{A} = (Q, \Sigma, \delta, \lambda, \rho)$ be a WFA and $R \in \mathbb{N}$. An **event function** is a tuple $\phi = (\phi_\lambda, \phi_\delta, \phi_\rho)$, where $\phi_\delta \colon Q \times \Sigma \times Q \to [R]_0$ and $\phi_\lambda, \phi_\rho \colon Q \to [R]_0$. For $\phi_f$ and $n \in [R]_0$, we define the

---

[4] The framework also supports an **at-least-$N$** variant (total $\mathcal{T}$-transitions $\geqslant N$; Thm. E.1), but we only consider the exact and at-least-once in the case studies (§5).

[5] Equivalently, $[\![\mathbb{K}]\!]_R$ is the monoid semialgebra over $\mathbb{K}$ generated by the cyclic monoid $\{c^0, c^1, \ldots, c^R\}$ with relation $c^R \cdot c = c^R$; under this presentation an element corresponds to a formal polynomial $\sum_{r=0}^R v_r c^r$, and $\circledtimes$ is polynomial multiplication modulo the relation. We discuss this alternative algebraic perspective in App. D.1.

**lifting function** $\mathcal{L}_f \colon \mathrm{dom}(f) \to [\![\mathbb{K}]\!]_R$

$$\mathcal{L}_f(x)_n \stackrel{\text{def}}{=} \begin{cases} f(x) & \text{if } n = \phi_f(x) \\ \mathbf{0} & \text{otherwise}. \end{cases} \tag{12}$$

Informally, the lifting function takes a weight in the original WFA and inserts it into the $\phi_f(x)^{\text{th}}$ position of the lifted weight in $[\![\mathbb{K}]\!]_R$; all other elements are $\mathbf{0}$.

**Definition 4.1** (Binning Automaton). *Let $\mathcal{A} = (Q, \Sigma, \delta, \lambda, \rho)$ be a WFA and $\phi = (\phi_\lambda, \phi_\delta, \phi_\rho)$ an event function on $\mathcal{A}$. The **binning automaton** of $\mathcal{A}$ over the $R^{th}$-order binning semiring with respect to $\mathbb{K}$ is the WFA $\mathcal{A}_\phi = (Q, \Sigma, \delta_\phi, \lambda_\phi, \rho_\phi)$, where*

$$\delta_\phi(q, \sigma, q') \stackrel{\text{def}}{=} \mathcal{L}_{\phi_\delta}\big(\delta(q, \sigma, q')\big), \tag{13}$$

$$\lambda_\phi(q) \stackrel{\text{def}}{=} \mathcal{L}_{\phi_\lambda}\big(\lambda(q)\big), \tag{14}$$

$$\rho_\phi(q) \stackrel{\text{def}}{=} \mathcal{L}_{\phi_\rho}\big(\rho(q)\big). \tag{15}$$

We can thus count the number of times an event $\phi$ occurs on a path $\boldsymbol{\pi} = q_0 \xrightarrow{a_1/w_1} \cdots \xrightarrow{a_m/w_m} q_m$ with

$$|\boldsymbol{\pi}|_\phi \stackrel{\text{def}}{=} \phi_\lambda(q_0) + \sum_{i=1}^m \phi_\delta(q_{i-1}, a_i, q_i) + \phi_\rho(q_m). \tag{16}$$

Similarly, for a multiset of paths $P = \{\boldsymbol{\pi}_k\}_{k=1}^K$, we write $|P|_\phi \stackrel{\text{def}}{=} \sum_{k=1}^K |\boldsymbol{\pi}_k|_\phi$. For path weights in $\mathcal{A}_\phi$, we use $\boldsymbol{w}_\phi(\boldsymbol{\pi}) \in [\![\mathbb{K}]\!]_R$ to denote the path weight function for $\boldsymbol{\pi} \in \Pi(\mathcal{A}_\phi)$. The next two results interpret the lifted weights.

**Theorem 4.1** (Path Weight Interpretation). *Let $\mathcal{A}$ be a WFSA over a semiring $(\mathbb{K}, \oplus, \otimes, \mathbf{0}, \mathbf{1})$, $R \in \mathbb{N}$, $\phi = (\phi_\lambda, \phi_\delta, \phi_\rho)$ an event function on $\mathcal{A}$, and $\mathcal{A}_\phi$ the corresponding binning automaton over $[\![\mathbb{K}]\!]_R$ (Definition 4.1). For any path $\boldsymbol{\pi} \in \Pi(\mathcal{A}_\phi)$, let $\boldsymbol{\pi}_\mathcal{A} \in \Pi(\mathcal{A})$ denote its base-automaton counterpart and $\boldsymbol{w}(\boldsymbol{\pi}_\mathcal{A}) \in \mathbb{K}$ its weight in $\mathcal{A}$. Then for any $n \in [R]_0$,*

$$\boldsymbol{w}_\phi(\boldsymbol{\pi})_n = \begin{cases} \boldsymbol{w}(\boldsymbol{\pi}_\mathcal{A}) & \text{if } n < R \text{ and } |\boldsymbol{\pi}|_\phi = n \\ \boldsymbol{w}(\boldsymbol{\pi}_\mathcal{A}) & \text{if } n = R \text{ and } |\boldsymbol{\pi}|_\phi \geqslant R \\ \mathbf{0} & \text{otherwise}. \end{cases} \tag{17}$$

*Proof.* See App. E. ∎

That is, if $|\boldsymbol{\pi}|_\phi < R$, the lifted weight places the base weight $\boldsymbol{w}(\boldsymbol{\pi}_\mathcal{A})$ at position $|\boldsymbol{\pi}|_\phi$, with $\mathbf{0}$ elsewhere. If $|\boldsymbol{\pi}|_\phi \geqslant R$, it places it at position $R$. For the remainder of this section we take $\mathcal{A}$ to be a PFA, i.e., the base semiring is the non-negative reals. Next, we formalize the intuition that summing path weights results in a vector that encodes the probabilities of sampling specific numbers of occurrences. To formalize this, let $\boldsymbol{\Pi}_\mathcal{A}(q)$ be a random variable over

paths starting from $q$ in the PFA $\mathcal{A}$, given by

$$\mathbb{P}(\boldsymbol{\Pi}_\mathcal{A}(q) = q \xrightarrow{a_1/w_1} \cdots \xrightarrow{a_M/w_M} q_M)$$
$$\stackrel{\text{def}}{=} \left(\prod_{m=1}^M w_m\right) \rho(q_M). \tag{18}$$

**Theorem 4.2** (Backward Weight Interpretation). *Under the conditions of Thm. 4.1, let $\boldsymbol{\Pi}_\mathcal{A}(q)$ denote a random path sampled from $\mathcal{A}$ starting at state $q$, and let $\boldsymbol{\beta}_\phi \stackrel{\text{def}}{=} \boldsymbol{\beta}_{\mathcal{A}_\phi} \colon Q \to [\![\mathbb{K}]\!]_R$ denote the backward-weight function (Eq. (3)) of $\mathcal{A}_\phi$. For any $q \in Q$ and $n \in [R]_0$,*

$$\boldsymbol{\beta}_\phi(q)_n = \begin{cases} \mathbb{P}(|\boldsymbol{\Pi}_\mathcal{A}(q)|_\phi = n) & \text{if } n < R \\ \mathbb{P}(|\boldsymbol{\Pi}_\mathcal{A}(q)|_\phi \geqslant R) & \text{if } n = R. \end{cases} \tag{19}$$

*Proof.* See App. E. ∎

That is, summing lifted path weights from $q$ yields the distribution over target counts—exactly the conditional we need to drive the sampler. Computing backward weights in a general PFA with cycles requires asteration, i.e., summation over the weights of infinitely many paths. We derive the closed-form solution for asteration in the binning semiring in App. D.2. The backward weights defined in Eq. (19) are the building block of the corpus sampler we develop next.

### 4.3. Constrained Sampling

With the backward weights $\boldsymbol{\beta}_\phi$ (Thm. 4.2) in hand, we sample a corpus $P = \{\boldsymbol{\pi}_k\}_{k=1}^K$ of $K$ paths from $\mathcal{A}$ subject to the corpus-level constraint $|P|_\phi = N$. The natural decomposition is two-phase: first sample the per-string counts $(n_1, \ldots, n_K)$ with $\sum_{k=1}^K n_k = N$, then sample each path with its target count. The two theorems below give the count distributions; `SamplePath` (Alg. 2) and `SampleCorpus` (Alg. 3) use them.

**Theorem 4.3** (Counting in a Set). *Under the conditions of Thm. 4.2, let $P = \{\boldsymbol{\pi}_k\}_{k=1}^K$ be a corpus of $K$ paths sampled i.i.d. from $\boldsymbol{\Pi}_\mathcal{A}(q_\iota)$, and let $\boldsymbol{Z} \stackrel{\text{def}}{=} \boldsymbol{\beta}_\phi(q_\iota) = \bigoplus_{\boldsymbol{\pi} \in \Pi(\mathcal{A}_\phi)} \boldsymbol{w}_\phi(\boldsymbol{\pi})$ be the sum of all path weights in $\mathcal{A}_\phi$. For any $n \in [R]_0$, we have*

$$(\boldsymbol{Z}^{\otimes K})_n = \begin{cases} \mathbb{P}(|P|_\phi = n) & \text{if } n < R \\ \mathbb{P}(|P|_\phi \geqslant R) & \text{if } n = R. \end{cases} \tag{20}$$

*Proof.* See App. E. ∎

**Theorem 4.4** (Sampling Path Event Counts). *Under the conditions of Thm. 4.3 (a corpus $P$ of $K$ i.i.d. paths from $\boldsymbol{\Pi}_\mathcal{A}(q_\iota)$), let $N \in \mathbb{N}$ be a target total count and set the*

**Algorithm 1** `Preprocess`: lifts $\mathcal{A}$ over $[\![\mathbb{K}]\!]_R$, computes the backward weights $\boldsymbol{\beta}_\phi$, stores each state's outgoing transitions in $\mathcal{E}$, and precomputes the per-action binning-weight table $\boldsymbol{W}$ used by `SamplePath`.

```
1  def Preprocess(A, φ, N):
2      # Lift to A_φ over [[K]]_R,  R = N + 1 (Def.4.1)
3      A_φ ← lift(A, φ, [[K]]_R, N+1)
4      β_φ ← backward_weights(A_φ)  # App. D.2
5      E(q) ← {q --a/w_e--> q′ ∈ δ} for q ∈ Q
6      # Precompute per-action binning weights from A_φ
7      for q ∈ Q:
8          W(q, EOS) ← ρ_φ(q)    # lifted accept weight
9          for q --a/w_e--> q′ ∈ E(q):
10             W(q, a) ← δ_φ(q, a, q′) ⊗ β_φ(q′)
11     return (β_φ, E, W)
```

**Algorithm 2** `SamplePath`: draws a single path from $\mathcal{A}$ with exactly $n$ target occurrences via per-step conditional sampling, looking up the preprocessed weight table $\boldsymbol{W}$ at the remaining budget $n - j$ at each step.

```
1  def SamplePath(A, φ, W, n, E):
2      q, j, π ← q_ι, 0, ()  # state, seen occ., path
3      while True:
4          # weights over E(q) ∪ {EOS} at budget n − j
5          w ← W(q, ·)_{n−j}
6          sample q --a/w_e--> q′ with Pr(·)∝w
7          if a = EOS: return π
8          π.append(q --a/w_e--> q′);  q, j ← q′, j + φ_δ(q, a, q′)
```

*binning semiring order to* $R = N + 1$. *For any* $k \in [K]$, $m \in [N]_0$, *and* $n \in [N - m]_0$,

$$\mathbb{P}(|\boldsymbol{\pi}_k|_\phi = n \mid |P|_\phi = N, |\{\boldsymbol{\pi}_j\}_{j<k}|_\phi = m)$$
$$= \frac{(\boldsymbol{Z}^{\circledast K-k})_{N-m-n}}{(\boldsymbol{Z}^{\circledast K-k+1})_{N-m}} \boldsymbol{Z}_n. \quad (21)$$

*Proof.* See App. E. ∎

Two analogous results in App. E cover related sampling variants. Thm. E.1 relaxes the corpus-level constraint from $|P|_\phi = N$ to $|P|_\phi \geqslant N$, returning the probability that path $\boldsymbol{\pi}_k$ contributes at least $n$ occurrences instead of exactly $n$. Thm. E.2 replaces the raw count by its per-path indicator, so the constraint becomes the *at-least-once* intervention.

**Constructing an efficient sampler.** The algorithm `SampleCorpus` first applies Thm. 4.4 to sample the per-string counts $(n_1, \ldots, n_K)$. Then, for each $k$ it calls `SamplePath` to draw a path with exactly $n_k$ target occurrences. `SamplePath` walks the original automaton state by

**Algorithm 3** `SampleCorpus`: draws a corpus of $K$ paths with corpus-level target count $N$ in two phases (per-string counts, then paths).

```
1   def SampleCorpus(A, φ, N, K):
2       β_φ, E, W ← Preprocess(A, φ, N)
3       Z, m ← β_φ(q_ι), 0
4       # Phase 1: per-string counts (Thm. 4.4)
5       for k in 1..K:
6           sample n_k with Pr(n_k=i)∝Z_i(Z^{⊗K−k})_{N−m−i}
7           m ← m + n_k
8       # Phase 2: paths conditioned on counts
9       for k in 1..K:
10          π_k ← SamplePath(A, φ, W, n_k, E)
11      return {π_k}_{k=1}^K
```

state, at each state sampling an outgoing transition or acceptance according to the probabilities conditioned on the number of occurrences remaining. We assume $\phi_\lambda(q_\iota) = 0$ throughout, i.e., interventions never occur on the initial-weight component. For any per-path target $n$, `SamplePath` returns a path distributed as $\mathbb{P}(\boldsymbol{\Pi}_\mathcal{A}(q_\iota) \mid |\boldsymbol{\Pi}_\mathcal{A}(q_\iota)|_\phi = n)$ as shown in the following theorem.

**Theorem 4.5** (Constrained Sampling). *Under the conditions of Thm. 4.4, fix $n \in [N]_0$ and define a lifted PFA $\mathcal{A}'$ over the state set $Q \times [n]_0$ (see App. E). Let $\boldsymbol{\Pi}_{\mathcal{A}'}(q_\iota)$ be a random path drawn from $\mathcal{A}'$. For any $\boldsymbol{\pi} \in \Pi(\mathcal{A}')$,*

$$\mathbb{P}(\boldsymbol{\Pi}_\mathcal{A}(q_\iota) = \boldsymbol{\pi}_\mathcal{A} \mid |\boldsymbol{\Pi}_\mathcal{A}(q_\iota)|_\phi = n) = \mathbb{P}(\boldsymbol{\Pi}_{\mathcal{A}'}(q_\iota) = \boldsymbol{\pi}).$$

*Proof.* See App. E. ∎

The *at-least* version (Thm. E.3 in App. E) samples a path conditioned on $|\boldsymbol{\Pi}_\mathcal{A}(q_\iota)|_\phi \geqslant n$ instead of $= n$. App. F.2 derives runtime bounds for all three algorithms. Let $K$ be the number of strings, $N$ the target count, and $\ell$ a bound on the sampled path length. `Preprocess` runs in $\mathcal{O}(|Q|^3 N^2)$ time, dominated by Lehmann's algorithm (Lehmann, 1977) on the binning-lifted automaton; each `SamplePath` call runs in $\mathcal{O}(\ell)$ with alias-table sampling (Walker, 1977; Vose, 1991); and the full `SampleCorpus` pipeline (one `Preprocess`, $K$ `SamplePath` calls, and the per-string count sampling) in $\mathcal{O}((|Q|^3 + K)N^2 + K\ell)$, or $\mathcal{O}((|Q|^3 + K)N \log N + K\ell)$ when the binning multiplication is computed with the FFT.

**Rejection sampling.** An alternative method for sampling under the property constraint is to draw a corpus and accept if it matches $\mathcal{P}$. It has an acceptance probability that decays exponentially in the gap between $N$ and the natural expected count of $\mathcal{T}$, so it is impractical for most events of interest; App. F.1 discusses naïve baselines in more detail.

## 5. Case Studies in Three Settings

We now compute and compare the correlational and causal estimates (Eq. (9b)) across three case studies: (i) the PARITY + star-free automaton of Fig. 1 with resampled weights (Fig. 2); (ii) a large random sample of 50-state PFAs (Fig. 5); and (iii) a fixed 40-state topology with resampled weights (Fig. 6).

**Sampling PFAs.** Case studies (ii) and (iii) sample each PFA of fixed size in two stages. *Topology:* for each source state $q \in Q$, we include each symbol $\sigma \in \Sigma$ independently with probability $\frac{1}{2}$ and assign each included symbol to a uniformly random target state $q' \in Q$. Each state is made accepting (nonzero final weight) independently with probability $\frac{3}{10}$. *Weights:* the outgoing transition weights at each state are drawn from a Dirichlet distribution. At accept states, the final weight is pinned at $\frac{3}{10}$ and the transition weights are rescaled to sum to $\frac{7}{10}$; at non-accept states the transition weights sum to 1. Full sampling details are given in App. H.

**Training.** Because we have access to the PFA that generated the data, each position supplies the full target distribution rather than a single sampled symbol. We therefore train against the summed per-position KL between the automaton's and the model's next-symbol distributions to get a more data-efficient signal than next-symbol cross-entropy:

$$
\begin{aligned}
\mathcal{L}(\boldsymbol{\sigma}) &= \sum_{t=1}^{|\boldsymbol{\sigma}|+1} \mathrm{D}_{\mathrm{KL}}\big(p_{\mathcal{A}}(\cdot \mid \boldsymbol{\sigma}_{<t}) \,\|\, p_{\boldsymbol{\theta}}(\cdot \mid \boldsymbol{\sigma}_{<t})\big) \\
&= \sum_{t=1}^{|\boldsymbol{\sigma}|+1} \sum_{\sigma \in \Sigma_{\mathrm{EOS}}} p_{\mathcal{A}}(\sigma \mid \boldsymbol{\sigma}_{<t}) \log \frac{p_{\mathcal{A}}(\sigma \mid \boldsymbol{\sigma}_{<t})}{p_{\boldsymbol{\theta}}(\sigma \mid \boldsymbol{\sigma}_{<t})},
\end{aligned}
\tag{22}
$$

where the sum runs to $|\boldsymbol{\sigma}| + 1$ to account for the full sequence along with termination. Across all three case studies we train both LSTM (Hochreiter and Schmidhuber, 1997) and transformer (Vaswani et al., 2017) architectures; full hyperparameters are in App. I.

**Decomposed Kullback–Leibler Divergence.** Each intervention targets a specific set of transitions (those leaving a chosen state, emitting a chosen symbol, or a single transition), so the measure of model fit must be localized to those transitions as well. However, the Kullback–Leibler (KL) divergence $\mathrm{D}_{\mathrm{KL}}(p_{\mathcal{A}} \,\|\, p_{\boldsymbol{\theta}})$ averages over *all* transitions and states, masking localized performance: a model that misses the targeted transitions can still look close overall by fitting well elsewhere. We therefore use a **decomposed KL divergence** that splits $\mathrm{D}_{\mathrm{KL}}(p_{\mathcal{A}} \,\|\, p_{\boldsymbol{\theta}})$ into per-state, per-transition, or per-symbol contributions, with only the targeted-transition term playing the role of $\mathrm{M}$ in the causal diagram (§3). Let $p_{\mathcal{A}}$ be the LM defined by a DPFA $\mathcal{A} = (Q, \Sigma, \delta, \lambda, \rho)$, and let $p_{\boldsymbol{\theta}}$ be the trained LM. Since

$\mathcal{A}$ is deterministic, the target state $q'$ is unique given $(q, \sigma)$ whenever $q \xrightarrow{\sigma/w} q' \in \delta$; we then write $p_{\mathcal{A}}(\sigma \mid q) = w$. Termination is governed by $p_{\mathcal{A}}(\mathrm{EOS} \mid q) = \rho(q)$, and $p_{\mathcal{A}}(\cdot \mid q)$ is a distribution over $\Sigma_{\mathrm{EOS}}$ at every state. We define

$$
p_{\pi}(q) \overset{\text{def}}{=} \sum_{\boldsymbol{\sigma} \in \Sigma^*} \overrightarrow{p}(\boldsymbol{\sigma}) \mathbb{1}\{\widehat{\delta}(\boldsymbol{\sigma}) = q\}, \tag{23a}
$$

$$
p_{\pi}(q, \sigma) \overset{\text{def}}{=} p_{\pi}(q)\, p_{\mathcal{A}}(\sigma \mid q), \tag{23b}
$$

$$
p_{\pi}(\sigma) \overset{\text{def}}{=} \sum_{q \in Q} p_{\pi}(q, \sigma). \tag{23c}
$$

The decompositions are as follows (derived in App. G):

$$
\mathrm{D}_{\mathrm{KL}}(p_{\mathcal{A}} \,\|\, p_{\boldsymbol{\theta}}) \tag{24a}
$$

$$
= \sum_{q \in Q} p_{\pi}(q) \underbrace{\sum_{\substack{\boldsymbol{\sigma} \in \Sigma^*, \\ \widehat{\delta}(\boldsymbol{\sigma})=q}} \frac{\overrightarrow{p}(\boldsymbol{\sigma})}{p_{\pi}(q)} \mathrm{D}_{\mathrm{KL}}\big(p_{\mathcal{A}}(\cdot \mid q) \,\|\, p_{\boldsymbol{\theta}}(\cdot \mid \boldsymbol{\sigma})\big)}_{\text{per-state contribution}} \tag{24b}
$$

$$
= \sum_{\substack{q \in Q, \\ \sigma \in \Sigma_{\mathrm{EOS}}}} p_{\pi}(q, \sigma) \underbrace{\sum_{\substack{\boldsymbol{\sigma} \in \Sigma^*, \\ \widehat{\delta}(\boldsymbol{\sigma})=q}} \frac{\overrightarrow{p}(\boldsymbol{\sigma})}{p_{\pi}(q)} \log \frac{p_{\mathcal{A}}(\sigma \mid q)}{p_{\boldsymbol{\theta}}(\sigma \mid \boldsymbol{\sigma})}}_{\text{per-transition contribution}} \tag{24c}
$$

$$
= \sum_{\sigma \in \Sigma_{\mathrm{EOS}}} p_{\pi}(\sigma) \underbrace{\sum_{\boldsymbol{\sigma} \in \Sigma^*} \frac{\overrightarrow{p}(\boldsymbol{\sigma}) p_{\mathcal{A}}(\sigma \mid \widehat{\delta}(\boldsymbol{\sigma}))}{p_{\pi}(\sigma)} \log \frac{p_{\mathcal{A}}(\sigma \mid \widehat{\delta}(\boldsymbol{\sigma}))}{p_{\boldsymbol{\theta}}(\sigma \mid \boldsymbol{\sigma})}}_{\text{per-symbol contribution}} \tag{24d}
$$

**PARITY + Star-Free Automaton.** We fix the topology of Fig. 1 and take $\mathcal{T}$ to be the transitions leaving a chosen state, sweeping the choice over $Q$ to read off the per-state decomposed KL. On it we sample 200 PFAs by drawing the outgoing transition weights at each state from a Dirichlet distribution. At the two accept states (ODD and REP in Fig. 1) the final (accept) weight is pinned at $\frac{1}{10}$ and the transition weights are rescaled to sum to $\frac{9}{10}$; at the other states the transition weights sum to 1. We use 80 of the 200 PFAs for the causal intervention (a single causal intervention yields many datapoints per PFA, while a correlational draw yields one per PFA) and all 200 for the correlational baseline. For each (PFA, intervention setting) pair we draw a 500-string corpus and train 10 LSTMs and 10 transformers; per architecture we report the run with the lowest total $\mathrm{D}_{\mathrm{KL}}(p_{\mathcal{A}} \| p_{\boldsymbol{\theta}})$, so that the architecture comparison is not confounded by training-run variance. We run both intervention types of §4 on this automaton, with exact-count results in Fig. 2 and at-least-once results in Fig. 7. Both show a clear causal–correlational gap, while the shapes differ.

**Aggregating Over Topologies.** We sample 952 random topologies of 50 states and 10 symbols as described above. On 97 of them we apply both the exact-count and at-least-once interventions, sampling 10 weight configurations per

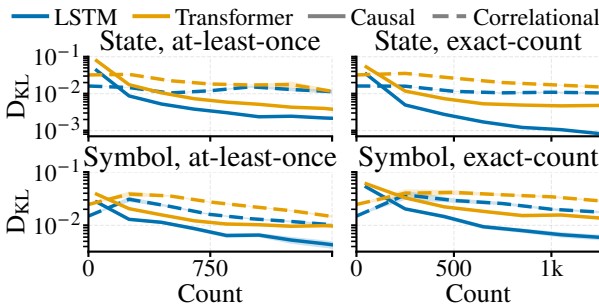

*Figure 5.* Monte Carlo estimates of the relation between learnability and the number of target occurrences over automata with 50 states and 10 symbols, for both state- and symbol-level interventions. Shaded areas represent one standard error.

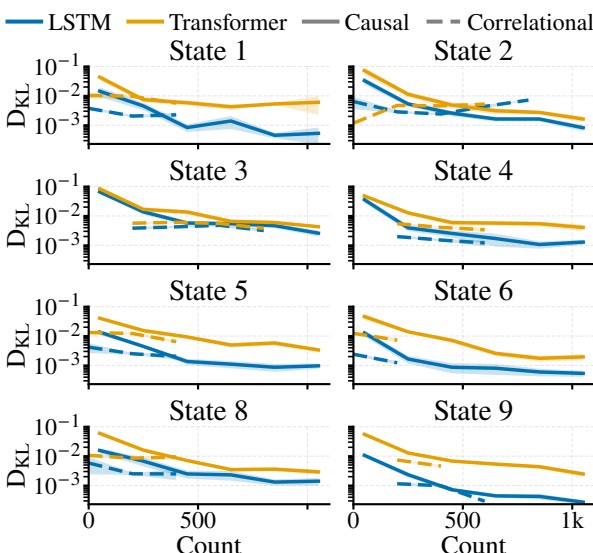

*Figure 6.* MCE of the Decomposed KL at randomly chosen states in a sampled automaton, with varying weight configurations. We see how correlational results can lead to misleading assumptions. Shaded areas represent one standard error.

topology. We draw a 500-string corpus per configuration for exact-count and a 2000-string corpus for at-least-once, and train one LSTM and one transformer on each. We compare two choices of $\mathcal{T}$: symbol-level, targeting transitions that emit a chosen symbol, and state-level, targeting transitions leaving a chosen state. Results for both properties appear in Fig. 5. Under both properties, the causal curves differ from the correlational baseline. At lower occurrence counts, the two curves also run in opposite directions: the causal symbol-level KL decreases as the target count grows, whereas the correlational KL first rises to a peak near 250 occurrences before falling. This reflects confounding: unconstrained sampling rarely yields a corpus with so few target occurrences, and the automata that do populate the low-count bins are easier to learn.

**Only Varying Weights.** We draw a single topology of 40 states and 10 symbols using the procedure above (so the topology is not hand-designed) and freeze it for the rest of the experiment. As in Fig. 2, we use a state-level property. On top of the frozen topology we sample 400 weight configurations for the correlational baseline and 10 weight configurations for the at-least-once intervention, sampling a 2000-string corpus per configuration. For each configuration we train five LSTMs and five transformers, visualising eight states in Fig. 6. Two findings extend the PARITY + starfree experiment (Fig. 1). First, a causal–correlational gap is visible at most states (note the log scale), and at low counts, it is approaching, or above, an order of magnitude at states 2, 6, and 8; this rules out the explanation that confounding is an artifact of a hand-designed toy task. Second, both the per-state learnability and the shape of the gap vary sharply within the same topology. State 3 has the highest LSTM end-point KL and is among the hardest for the Transformer; state 1 is the hardest for the Transformer; state 9 is the easiest for the LSTM. The gap itself takes different shapes: at state 2 (both architectures) and state 8 (Transformer), the causal curve descends sharply while the correlational curve

stays roughly flat or even rises; at state 4, the two curves run nearly parallel throughout; at state 3, the causal curve descends to converge with a flat correlational baseline at the right edge of the correlational range. These are state-level variations within the same topology, making confounding a state-level rather than topology-level phenomenon.

## 6. Conclusion

We have made the case that language models trained on formal languages should be thought of as multi-task learners, much like modern LMs. This means inter-task confounders must be controlled when studying how an architecture learns a *given* task. We present the tools needed to do so—the binning semiring and the sampling algorithms based on it. Our case studies support this story: removing the confounders changes the learnability curves. For example, the non-monotonic correlational trend at low occurrence counts in Fig. 5 disappears under causal sampling. Collectively, the apparent effect of a property on learnability depends on the sampling procedure: correlational filtering of unintervened corpora confounds the property with the structural features it co-varies with, so the curves can reflect the sampler rather than the property itself. This matters for any study comparing architectures, characterizing scaling behavior, or attributing difficulty to specific structural features; we hope the methodology developed here offers a useful tool for such work. The framework extends naturally to richer automata (e.g., pushdown automata for context-free languages). Limitations are given in App. A.

## Impact Statement

This paper proposes a causal alternative to correlational evaluation of language-model learnability on formal tasks. Its impact is mediated through research practice: adopting these methods would change how learnability claims are reported and may require revisiting prior correlational findings about how language properties affect learnability.

## Acknowledgements

VS and JV were supported by the Pioneer Centre for AI, DNRF grant number P1.

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

# Contents of the Appendix

## A. Limitations

We focus on PFA-induced settings where properties are exactly countable; applying the framework to natural language would require approximate labels for target phenomena. The computational cost of our sampling algorithm grows with the size of the automaton (closed-form bounds in App. F.2). Each architecture is trained at a single fixed size, without scaling or hyperparameter sweeps. We note that our experiments and results only concern the causal procedure as such, including the controlled sampling and evaluation. We are not claiming to present a thorough empirical study of any architecture or automaton; the choice of LSTM and transformer architectures, and of PARITY-based automata, is illustrative; the framework extends to other architectures and tasks without modification.

## B. Related Work

Much work has used formal languages to probe neural networks (Cleeremans et al., 1989; Jacobsson, 2005; Valvoda et al., 2022; Svete et al., 2024b; Borenstein et al., 2024, *inter alia*). A central empirical tool is synthetic data: SCAN-style setups examine compositional generalization (Lake and Baroni, 2018; Bastings et al., 2018; Ruis et al., 2020), and $k$-Dyck languages probe nested structure (Weiss et al., 2018; Suzgun et al., 2019; Bhattamishra et al., 2020; Hewitt et al., 2020). Recent work also analyzes inductive biases relative to the Chomsky hierarchy (Delétang et al., 2023; Butoi et al., 2025) and studies whole *classes* of languages rather than single datasets (Valvoda et al., 2022; Borenstein et al., 2024), continuing a line of grammatical inference research (Jacobsson, 2005). Closest to our controlled sampling is MLRegTest (Van der Poel et al., 2025), a benchmark built by intersecting an unweighted DFA with a length-constraining automaton and sampling uniformly over the resulting paths. Our sampler instead draws from the exact conditional distribution induced by a *probabilistic* FA, conditioned on transition- or symbol-level occurrence counts rather than string length. Complementing these empirical traditions, theory investigates the representational capacity and internal mechanisms of neural LMs and architectures, especially transformers (Merrill, 2023; Strobl et al., 2023; Merrill, 2019; Merrill et al., 2020; Liu et al., 2023, *inter alia*). Like our study, this line probes how language complexity interacts with the neural network architecture, but yields bounds under idealized assumptions rather than measurements of the actual learning dynamics under noisy training. By contrast, there has been relatively little emphasis on *causal* approaches to LM behavior. Causal investigation with natural language is difficult—requiring complex taxonomies (Chen et al., 2024) or targeted neuron-level interventions (Vig et al., 2020; Finlayson et al., 2021). Methodologically closer to our setup are *data-level* interventions on training data, such as counterfactual data augmentation (Kaushik et al., 2020), and per-task scaling studies (Hoffmann et al., 2022); the binning semiring localizes both ideas from the corpus level to the level of designated automaton substructures. The same conditional distribution could in principle be obtained by rejection sampling from the unconstrained PFA, but its acceptance probability decays exponentially in the gap to the natural count (§4); our sampler realizes it in polynomial time. We note that our focus is *learnability under standard, fixed-size training setups and natural data*, not strict expressivity.

## C. Markov Kernels

The body of the paper (§2) defines a graphical causal model using a set of conditional distributions, one per variable, and treats each one as a conditional density. For continuous parent variables, conditional densities exist whenever the joint is absolutely continuous with respect to a product reference measure; the fully general object that handles measure-zero conditioning events is a Markov kernel, which we define here.

A $\sigma$-**algebra** on a set $\mathcal{X}$ is a collection $\mathfrak{X}$ of subsets of $\mathcal{X}$ that contains $\mathcal{X}$ and is closed under complementation and countable unions. A **measurable space** is a pair $(\mathcal{X}, \mathfrak{X})$ where $\mathcal{X}$ is a set and $\mathfrak{X}$ is a $\sigma$-algebra on $\mathcal{X}$. A **measure** on $(\mathcal{X}, \mathfrak{X})$ is a function $\mu : \mathfrak{X} \to [0, \infty]$ such that $\mu(\varnothing) = 0$ and $\mu$ is countably additive: for any countable collection $\{x_i\}_{i \in \mathbb{N}}$ of disjoint sets in $\mathfrak{X}$,

$$\mu \left( \bigcup_{i \in \mathbb{N}} x_i \right) = \sum_{i \in \mathbb{N}} \mu(x_i). \tag{25}$$

A **probability measure** on $(\mathcal{X}, \mathfrak{X})$ is a measure $\mathbb{P}$ such that $\mathbb{P}(\mathcal{X}) = 1$.

**Notation.** We write variables for sets in uppercase calligraphic font, e.g., $\mathcal{X}$ and $\mathcal{Y}$, and variables for $\sigma$-algebras in uppercase Fraktur, e.g., $\mathfrak{X}$ and $\mathfrak{Y}$. We write measurable spaces as, e.g., $(\mathcal{X}, \mathfrak{X})$.

A **Markov kernel**[6] from $\mathcal{X}$ to $\mathcal{Y}$ is a function $\kappa \colon \mathcal{X} \times \mathfrak{Y} \to [0, 1]$ such that *(i)* for each $x \in \mathcal{X}$, $\kappa(x, \cdot)$ is a probability measure on $(\mathcal{Y}, \mathfrak{Y})$, and *(ii)* for each $Y \in \mathfrak{Y}$, $\kappa(\cdot, Y)$ is $\mathfrak{X}$-measurable. We say that a Markov kernel is **trivial** if $\mathcal{X} = \varnothing$. In this case, $\kappa$ corresponds to a probability measure on $(\mathcal{Y}, \mathfrak{Y})$.

## D. The Binning Semiring

### D.1. Monoid Semialgebras

We record the monoid-semialgebra perspective for completeness: the binning semiring is a particular monoid semialgebra (over the cyclic monoid below), offering an alternative algebraic framing to the operational definition in §4.1. The proofs that follow do not directly rely on this framing, but it makes explicit why $[\![\mathbb{K}]\!]_R$ inherits standard semiring structure.

**Definition D.1.** *A **cyclic monoid** is a monoid with a single generator $c$ (every element is a power $c^i$ for some $i \geqslant 0$) and an identity element. We denote by $C_{R,1}$ the monoid $C_{R,1} \overset{\text{def}}{=} \{c^0, c^1, \ldots, c^R\}$ where $c^0$ is the identity. Multiplication is defined for $0 \leqslant i, j \leqslant R$ as*

$$c^i \cdot c^j \overset{\text{def}}{=} \begin{cases} c^{i+j} & \text{if } i + j < R \\ c^R & \text{otherwise}. \end{cases} \tag{26}$$

*In particular, $c^R \cdot c^1 = c^R$.*

**Definition D.2.** *Given a finite monoid $M$ and a (semi)ring $(\mathbb{K}, \oplus, \otimes, \mathbf{0}, \mathbf{1})$, the **monoid semialgebra** $\mathbb{K}[M]$ consists of all formal sums*

$$f = \sum_{m \in M} a_m m \tag{27}$$

*where $a_m \in \mathbb{K}$, with point-wise addition. The outer $\sum$ and the juxtaposition $a_m m$ are formal-sum notation rather than ring operations; the ring structure $(\oplus, \otimes)$ enters only when the product below is evaluated. The product in $\mathbb{K}[M]$ is given by a convolution:*

$$\left( \sum_{m \in M} a_m m \right) \cdot \left( \sum_{n \in M} b_n n \right) = \sum_{m' \in M} \left( \sum_{\substack{m,n \in M \\ m \cdot n = m'}} a_m b_n \right) m' \tag{28}$$

An element $\sum_{m \in M} a_m m$ of $\mathbb{K}[M]$ is the monoid analogue of a polynomial $\sum_{i \geqslant 0} a_i x^i$ over $\mathbb{K}$, with the indeterminate $x$ replaced by elements of $M$.

Applying Definition D.2 to Definition D.1, we can see that an element of $[\![\mathbb{K}]\!]_R$ can be written uniquely as $\sum_{i=0}^{R} a_i c^i$ where $a_i, b_i \in \mathbb{K}$ (with $b_i$ the analogous coefficients of a second such element) and the convolution product is given by

$$\left( \sum_{i=0}^{R} a_i c^i \right) \cdot \left( \sum_{i=0}^{R} b_i c^i \right) = \sum_{i=0}^{R-1} \left( \sum_{j=0}^{i} a_j b_{i-j} \right) c^i + \left( \sum_{\substack{0 \leqslant i,j \leqslant R \\ i+j \geqslant R}} a_i b_j \right) c^R \tag{29}$$

### D.2. A Closed-form Solution for the Kleene Star

Computing backward weights in a general PFA with cycles requires summation over infinitely many paths. This can be done efficiently with *asteration*—taking the Kleene closure of the set of paths. Backward weights of the (lifted) automaton are computed by Lehmann's algorithm (Lehmann, 1977) (see App. F.2 for runtime), which invokes the element-wise Kleene-star formula derived below as a subroutine. We derive the closed-form Kleene-star (asteration) operator for binning automata applied to PFAs and characterize when the solution is unique.

**Lemma D.3** (Arden's rule in complete semirings). *Let $(\mathbb{K}, \oplus, \otimes, \mathbf{0}, \mathbf{1})$ be a complete semiring. For all $a, b \in \mathbb{K}$, the equation*

$$x = (a \otimes x) \oplus b \tag{30}$$

*has $x = a^* \otimes b$ as a solution.*

---

[6]A Markov kernel is a measure-theoretic surrogate for conditional probability that avoids the problem of conditioning on a measure-zero event: the elementary formula $\mathbb{P}(Y \in Y \mid X = x) = \mathbb{P}(X = x, Y \in Y)/\mathbb{P}(X = x)$ is ill-defined when $\mathbb{P}(X = x) = 0$, e.g., for continuous X. The map $x \mapsto \kappa(x, \cdot)$ is well-defined at every $x$ by construction.

*Proof.* By Kuich (1997, Thm. 2.2(ii)), $a^* = \mathbf{1} \oplus (a \otimes a^*)$. Therefore

$$
\begin{aligned}
(a \otimes (a^* \otimes b)) \oplus b &= (a \otimes a^*) \otimes b \oplus b \\
&= ((a \otimes a^*) \oplus \mathbf{1}) \otimes b \\
&= a^* \otimes b.
\end{aligned}
\tag{31}
$$

∎

**Proposition D.4** (Coefficientwise Kleene equations in the binning semiring). *Let $[\![\mathbb{K}]\!]_R$ be the Rth-order binning semiring and $\boldsymbol{v} \in [\![\mathbb{K}]\!]_R$. Write $\boldsymbol{v}^* \overset{\text{def}}{=} \bigoplus_{i=0}^{\infty} \boldsymbol{v}^{\otimes i}$. For $0 \leqslant n < R$,*

$$
(\boldsymbol{v}^*)_n = v_0^* \otimes \left( \textcircled{1}_n \oplus \bigoplus_{j=1}^{n} v_j \otimes (\boldsymbol{v}^*)_{n-j} \right).
\tag{32}
$$

*For the overflow bin,*

$$
(\boldsymbol{v}^*)_R = \left( \bigoplus_{n=0}^{R} v_n \right)^* \otimes \left( \bigoplus_{\substack{n,j=0 \\ j \neq R \\ n+j \geqslant R}}^{R} v_n \otimes (\boldsymbol{v}^*)_j \right).
\tag{33}
$$

*Proof.* For $0 \leqslant n < R$,

$$
\begin{aligned}
(\boldsymbol{v}^*)_n = \left( \bigoplus_{k=0}^{\infty} \boldsymbol{v}^{\otimes k} \right)_n &= \textcircled{1}_n \oplus (\boldsymbol{v} \otimes \boldsymbol{v}^*)_n \\
&= \textcircled{1}_n \oplus \bigoplus_{j=0}^{n} v_j \otimes (\boldsymbol{v}^*)_{n-j} \\
&= v_0 \otimes (\boldsymbol{v}^*)_n \oplus \left( \textcircled{1}_n \oplus \bigoplus_{j=1}^{n} v_j \otimes (\boldsymbol{v}^*)_{n-j} \right).
\end{aligned}
\tag{34}
$$

Apply Lemma D.3 with $a = v_0$ to obtain Eq. (32). For $n = R$,

$$
\begin{aligned}
(\boldsymbol{v}^*)_R = \textcircled{1}_R \oplus (\boldsymbol{v} \otimes \boldsymbol{v}^*)_R &= \textcircled{1}_R \oplus \bigoplus_{\substack{i,j=0 \\ i+j \geqslant R}}^{R} v_i \otimes (\boldsymbol{v}^*)_j \\
&= \left( \bigoplus_{i=0}^{R} v_i \right) \otimes (\boldsymbol{v}^*)_R \oplus \bigoplus_{\substack{i,j=0 \\ j \neq R, i+j \geqslant R}}^{R} v_i \otimes (\boldsymbol{v}^*)_j,
\end{aligned}
\tag{35}
$$

and Lemma D.3 with $a = \bigoplus_{i=0}^{R} v_i$ yields Eq. (33). ∎

**Lemma D.5** (Uniqueness in the binning semiring). *Let $(\mathbb{R}_{\geqslant 0}, +, \cdot, 0, 1)$ be the non-negative real semiring, and let $[\![\mathbb{K}]\!]_R$ be the binning semiring of order R over $\mathbb{R}_{\geqslant 0}$. For any $v, w \in [\![\mathbb{K}]\!]_R$ with $v$ arising from a PFA binning automaton and $\sum_{i=0}^{R} v_i < 1$ (a strict inequality, which implies $v_0 < 1$ and holds whenever the self-loop is not absorbing), the equation*

$$
x = (v \otimes x) \oplus w
\tag{36}
$$

*has a unique solution.*

*Proof.* For $0 \leqslant n < R$,

$$
x_n = v_0 \otimes x_n \oplus \bigoplus_{j=1}^{n} v_j \otimes x_{n-j} \oplus w_n.
\tag{37}
$$

Working in $\mathbb{R}_{>0}$ and using $v_0 < 1$,

$$
x_n = \frac{1}{1 - v_0} \cdot \left( \bigoplus_{j=1}^{n} v_j \otimes x_{n-j} \oplus w_n \right),
\tag{38}
$$

which uniquely determines $x_n$ by strong induction and preserves non-negativity. For $n = R$, the binning-semiring convolution at the overflow bin (Eq. (11)) gives

$$
x_R = \bigoplus_{\substack{i+j \geqslant R \\ 0 \leqslant i,j \leqslant R}} v_i \otimes x_j \oplus w_R.
\tag{39}
$$

The pairs $(i, j) = (i, R)$ for $i \in \{0, \dots, R\}$ each satisfy $i + R \geqslant R$, so the coefficient of $x_R$ on the right is $\bigoplus_{i=0}^{R} v_i$. Collecting these on the left yields

$$\left(1 - \bigoplus_{i=0}^{R} v_i\right) x_R = \bigoplus_{\substack{i+j \geqslant R \\ 0 \leqslant i \leqslant R, \; 0 \leqslant j < R}} v_i \otimes x_j \oplus w_R. \tag{40}$$

The right-hand side depends only on $x_0, \dots, x_{R-1}$, which were already uniquely determined by the strong induction above. Since $\bigoplus_{i=0}^{R} v_i < 1$, the coefficient on the left is positive, so $x_R$ is uniquely determined and non-negative. $\blacksquare$

**Corollary D.6** (Uniqueness of the closed form in the PFA/binning setting). *Under the conditions of Prop. D.4 and Lemma D.5, the solution given by Eqs.* (32) *and* (33) *is unique.*

# E. Proofs

**Theorem 4.1** (Path Weight Interpretation). *Let $\mathcal{A}$ be a WFSA over a semiring $(\mathbb{K}, \oplus, \otimes, \mathbf{0}, \mathbf{1})$, $R \in \mathbb{N}$, $\phi = (\phi_\lambda, \phi_\delta, \phi_\rho)$ an event function on $\mathcal{A}$, and $\mathcal{A}_\phi$ the corresponding binning automaton over $[\![\mathbb{K}]\!]_R$ (Definition 4.1). For any path $\boldsymbol{\pi} \in \Pi(\mathcal{A}_\phi)$, let $\boldsymbol{\pi}_\mathcal{A} \in \Pi(\mathcal{A})$ denote its base-automaton counterpart and $\boldsymbol{w}(\boldsymbol{\pi}_\mathcal{A}) \in \mathbb{K}$ its weight in $\mathcal{A}$. Then for any $n \in [R]_0$,*

$$\boldsymbol{w}_\phi(\boldsymbol{\pi})_n = \begin{cases} \boldsymbol{w}(\boldsymbol{\pi}_\mathcal{A}) & \textbf{if } n < R \text{ and } |\boldsymbol{\pi}|_\phi = n \\ \boldsymbol{w}(\boldsymbol{\pi}_\mathcal{A}) & \textbf{if } n = R \text{ and } |\boldsymbol{\pi}|_\phi \geqslant R \\ \mathbf{0} & \textbf{otherwise}. \end{cases} \tag{17}$$

*Proof.* Recall that $\phi = (\phi_\lambda, \phi_\delta, \phi_\rho)$ assigns event counts to $\mathcal{A}$'s initial, transition, and final weight functions $(\lambda, \delta, \rho)$ respectively. For a path $\boldsymbol{\pi} = q_0 \xrightarrow{a_1/\boldsymbol{w}_1} q_1 \cdots q_{m-1} \xrightarrow{a_m/\boldsymbol{w}_m} q_m \in \Pi(\mathcal{A}_\phi)$, the lifted (outer) path weight factorizes as $\boldsymbol{w}_\phi(\boldsymbol{\pi}) = \lambda_\phi(q_0) \otimes \overline{\boldsymbol{w}}_\phi(\boldsymbol{\pi}) \otimes \rho_\phi(q_m)$, where $\lambda_\phi(q_0), \rho_\phi(q_m) \in [\![\mathbb{K}]\!]_R$ are the lifted init/final weights (vectors with a single nonzero coordinate, at $\phi_\lambda(q_0)$ and $\phi_\rho(q_m)$ respectively), $\overline{\boldsymbol{w}}_\phi(\boldsymbol{\pi}) = \boldsymbol{w}_1 \otimes \cdots \otimes \boldsymbol{w}_m$ is the lifted inner weight with $\boldsymbol{w}_t = \mathcal{L}_{\phi_\delta}(q_{t-1} \xrightarrow{a_t/\boldsymbol{w}_t} q_t)$, and $\otimes$ is the binning semiring's convolution (combined counts $\geqslant R$ collapse to bin $R$).

We first establish, by induction on $|\boldsymbol{\pi}|$, the inner-weight version of the claim. Write $|\boldsymbol{\pi}|_\phi^\delta \overset{\text{def}}{=} \sum_{t=1}^{|\boldsymbol{\pi}|} \phi_\delta(q_{t-1} \xrightarrow{a_t/\boldsymbol{w}_t} q_t)$ for the transition-only feature sum, so that $|\boldsymbol{\pi}|_\phi = |\boldsymbol{\pi}|_\phi^\delta + \phi_\lambda(q_0) + \phi_\rho(q_{|\boldsymbol{\pi}|})$ by definition. For any $n \in [R]_0$,

$$\overline{\boldsymbol{w}}_\phi(\boldsymbol{\pi})_n = \begin{cases} \overline{\boldsymbol{w}}(\boldsymbol{\pi}_\mathcal{A}) & n < R \text{ and } \sum_{t=1}^{|\boldsymbol{\pi}|} \phi_\delta(q_{t-1} \xrightarrow{a_t/\boldsymbol{w}_t} q_t) = n, \\ \overline{\boldsymbol{w}}(\boldsymbol{\pi}_\mathcal{A}) & n = R \text{ and } \sum_{t=1}^{|\boldsymbol{\pi}|} \phi_\delta(q_{t-1} \xrightarrow{a_t/\boldsymbol{w}_t} q_t) \geqslant R, \\ \mathbf{0} & \textbf{otherwise}. \end{cases} \tag{41}$$

If $|\boldsymbol{\pi}| = 1$, write $\boldsymbol{\pi} = q_0 \xrightarrow{a_1/\boldsymbol{w}_1} q_1$, so $\boldsymbol{\pi}_\mathcal{A} = q_0 \xrightarrow{a_1/w_1} q_1$ and $\overline{\boldsymbol{w}}(\boldsymbol{\pi}_\mathcal{A}) = w_1$. Then for each $n \in [R]_0$,

$$\overline{\boldsymbol{w}}_\phi(\boldsymbol{\pi})_n = \left(\mathcal{L}_{\phi_\delta}(q_0 \xrightarrow{a_1/w_1} q_1)\right)_n = \begin{cases} \overline{\boldsymbol{w}}(\boldsymbol{\pi}_\mathcal{A}) & n = \phi_\delta(q_0 \xrightarrow{a_1/w_1} q_1) = |\boldsymbol{\pi}|_\phi^\delta, \\ \mathbf{0} & \textbf{otherwise}, \end{cases}$$

matching Eq. (41).

Suppose now that Eq. (41) holds for all paths $\boldsymbol{\pi}$ with $|\boldsymbol{\pi}| < m$. Let $\boldsymbol{\pi} = q_0 \xrightarrow{a_1/\boldsymbol{w}_1} q_1 \cdots \xrightarrow{a_m/\boldsymbol{w}_m} q_m$ be a path of length $m$, and $\boldsymbol{\pi}' = q_0 \xrightarrow{a_1/\boldsymbol{w}_1} q_1 \cdots \xrightarrow{a_{m-1}/\boldsymbol{w}_{m-1}} q_{m-1}$ be the sub-path of the first $m-1$ transitions. We compute for $n \in [R-1]_0$

$$\overline{\boldsymbol{w}}_\phi(\boldsymbol{\pi})_n = (\overline{\boldsymbol{w}}_\phi(\boldsymbol{\pi}') \otimes \boldsymbol{w}_m)_n \tag{42a}$$

$$= \sum_{j=0}^{n} \overline{\boldsymbol{w}}_\phi(\boldsymbol{\pi}')_j \otimes (\boldsymbol{w}_m)_{n-j} \tag{42b}$$

$$= \overline{\boldsymbol{w}}_\phi(\boldsymbol{\pi}')_{|\boldsymbol{\pi}'|_\phi^\delta} \otimes (\boldsymbol{w}_m)_{n-|\boldsymbol{\pi}'|_\phi^\delta} \qquad \text{(Inductive hypothesis (with } |\boldsymbol{\pi}'|_\phi^\delta < R \text{ since } n < R), \tag{42c}$$

$$= \begin{cases} \overline{\boldsymbol{w}}(\boldsymbol{\pi}') \otimes w_m = \overline{\boldsymbol{w}}(\boldsymbol{\pi}) & \textbf{if } n = |\boldsymbol{\pi}'|_\phi^\delta + \phi_\delta(q_{m-1} \xrightarrow{a_m/w_m} q_m) \\ \mathbf{0} & \textbf{otherwise} \end{cases}. \tag{42d}$$

This completes the proof for coefficients $n < R$. For the remaining case, we derive:

$$\overline{\boldsymbol{w}}_\phi(\boldsymbol{\pi})_R = (\overline{\boldsymbol{w}}_\phi(\boldsymbol{\pi}') \otimes \boldsymbol{w}_m)_R \tag{43a}$$

$$= \sum_{\substack{i,j=0 \\ i+j \geqslant R}}^{R} \overline{\boldsymbol{w}}_\phi(\boldsymbol{\pi}')_i \otimes (\boldsymbol{w}_m)_j \tag{43b}$$

$$= \begin{cases} \overline{\boldsymbol{w}}_\phi(\boldsymbol{\pi}')_{\min(|\boldsymbol{\pi}'|_\phi^\delta, R)} \otimes (\boldsymbol{w}_m)_{\phi_\delta(q_{m-1} \xrightarrow{a_m/w_m} q_m)} & \textbf{if } |\boldsymbol{\pi}'|_\phi^\delta + \phi_\delta(q_{m-1} \xrightarrow{a_m/w_m} q_m) \geqslant R \\ \mathbf{0} & \textbf{otherwise} \end{cases} \tag{43c}$$

(Inductive hypothesis, 43c)

$$= \begin{cases} \overline{\boldsymbol{w}}(\boldsymbol{\pi}') \otimes w_m = \overline{\boldsymbol{w}}(\boldsymbol{\pi}) & \textbf{if } |\boldsymbol{\pi}'|_\phi^\delta + \phi_\delta(q_{m-1} \xrightarrow{a_m/w_m} q_m) \geqslant R \\ \mathbf{0} & \textbf{otherwise} \end{cases}. \tag{43d}$$

This establishes Eq. (41). To conclude, convolve with the lifted init and final weights. Since $\lambda_\phi(q_0) \in [\![\mathbb{K}]\!]_R$ has its unique nonzero coordinate at index $\phi_\lambda(q_0)$ with value $\lambda(q_0)$, and similarly $\rho_\phi(q_m)$ at index $\phi_\rho(q_m)$ with value $\rho(q_m)$, the convolution $\lambda_\phi(q_0) \otimes \overline{\boldsymbol{w}}_\phi(\boldsymbol{\pi}) \otimes \rho_\phi(q_m)$ shifts the supported coordinate by $\phi_\lambda(q_0) + \phi_\rho(q_m)$, giving $\boldsymbol{w}_\phi(\boldsymbol{\pi})_n = \boldsymbol{w}(\boldsymbol{\pi}_\mathcal{A})$ at $n = \sum_{t=1}^{|\boldsymbol{\pi}|} \phi_\delta(\ldots) + \phi_\lambda(q_0) + \phi_\rho(q_m) = |\boldsymbol{\pi}|_\phi$ (with the overflow sub-case at $n = R$ when $|\boldsymbol{\pi}|_\phi \geqslant R$), and $\mathbf{0}$ elsewhere. ∎

**Theorem 4.2** (Backward Weight Interpretation). *Under the conditions of Thm. 4.1, let $\mathbf{\Pi}_\mathcal{A}(q)$ denote a random path sampled from $\mathcal{A}$ starting at state $q$, and let $\boldsymbol{\beta}_\phi \overset{\text{def}}{=} \boldsymbol{\beta}_{\mathcal{A}_\phi} : Q \to [\![\mathbb{K}]\!]_R$ denote the backward-weight function (Eq. (3)) of $\mathcal{A}_\phi$. For any $q \in Q$ and $n \in [R]_0$,*

$$\boldsymbol{\beta}_\phi(q)_n = \begin{cases} \mathbb{P}(|\mathbf{\Pi}_\mathcal{A}(q)|_\phi = n) & \textbf{if } n < R \\ \mathbb{P}(|\mathbf{\Pi}_\mathcal{A}(q)|_\phi \geqslant R) & \textbf{if } n = R. \end{cases} \tag{19}$$

*Proof.* Paths in $\mathcal{A}_\phi$ and $\mathcal{A}$ correspond bijectively via $\boldsymbol{\pi} \leftrightarrow \boldsymbol{\pi}_\mathcal{A}$ (see, e.g., Thm. 4.1). By Thm. 4.1, the lifted weight $\boldsymbol{w}_\phi(\boldsymbol{\pi})$ has its single nonzero coordinate at index $\min(|\boldsymbol{\pi}|_\phi, R)$, where it equals $\boldsymbol{w}(\boldsymbol{\pi}_\mathcal{A})$. Since $\mathcal{A}$ is a PFA, this equals the probability $\mathbb{P}(\mathbf{\Pi}_\mathcal{A}(q) = \boldsymbol{\pi}_\mathcal{A})$ assigned to $\boldsymbol{\pi}_\mathcal{A}$ in Eq. (18).

For $n \in [R-1]_0$,

$$\mathbb{P}(|\mathbf{\Pi}_\mathcal{A}(q)|_\phi = n) = \sum_{\substack{\boldsymbol{\pi} \in \Pi(\mathcal{A}_\phi) \\ |\boldsymbol{\pi}|_\phi = n}} \mathbb{P}(\mathbf{\Pi}_\mathcal{A}(q) = \boldsymbol{\pi}_\mathcal{A}) \tag{44a}$$

$$= \sum_{\substack{\boldsymbol{\pi} \in \Pi(\mathcal{A}_\phi) \\ |\boldsymbol{\pi}|_\phi = n}} \boldsymbol{w}(\boldsymbol{\pi}_\mathcal{A}) \qquad (\mathcal{A} \text{ is a PFA, 44b})$$

$$= \sum_{\substack{\boldsymbol{\pi} \in \Pi(\mathcal{A}_\phi) \\ |\boldsymbol{\pi}|_\phi = n}} \boldsymbol{w}_\phi(\boldsymbol{\pi})_n \qquad (\text{Thm. 4.1, with } |\boldsymbol{\pi}|_\phi = n < R, \text{44c})$$

$$= \sum_{\boldsymbol{\pi} \in \Pi(\mathcal{A}_\phi)} \boldsymbol{w}_\phi(\boldsymbol{\pi})_n \qquad (\text{Thm. 4.1: coordinate } n \text{ vanishes when } |\boldsymbol{\pi}|_\phi \neq n, \text{44d})$$

$$= \boldsymbol{\beta}_\phi(q)_n. \qquad (\text{Definition of } \boldsymbol{\beta}_\phi, \text{44e})$$

For the overflow coefficient $n = R$, the same chain applies with the at-least-$R$ branch of Thm. 4.1:

$$\mathbb{P}(|\mathbf{\Pi}_{\mathcal{A}}(q)|_\phi \geqslant R) = \sum_{\substack{\boldsymbol{\pi} \in \Pi(\mathcal{A}_\phi) \\ |\boldsymbol{\pi}|_\phi \geqslant R}} \mathbb{P}(\mathbf{\Pi}_{\mathcal{A}}(q) = \boldsymbol{\pi}_{\mathcal{A}}) \tag{45a}$$

$$= \sum_{\substack{\boldsymbol{\pi} \in \Pi(\mathcal{A}_\phi) \\ |\boldsymbol{\pi}|_\phi \geqslant R}} \boldsymbol{w}(\boldsymbol{\pi}_{\mathcal{A}}) \qquad\qquad (\mathcal{A} \text{ is a PFA, } 45b)$$

$$= \sum_{\substack{\boldsymbol{\pi} \in \Pi(\mathcal{A}_\phi) \\ |\boldsymbol{\pi}|_\phi \geqslant R}} \boldsymbol{w}_\phi(\boldsymbol{\pi})_R \qquad\qquad (\text{Thm. 4.1, overflow sub-case, } 45c)$$

$$= \sum_{\boldsymbol{\pi} \in \Pi(\mathcal{A}_\phi)} \boldsymbol{w}_\phi(\boldsymbol{\pi})_R \qquad\qquad (\text{Thm. 4.1: coordinate } R \text{ vanishes when } |\boldsymbol{\pi}|_\phi < R, 45d)$$

$$= \boldsymbol{\beta}_\phi(q)_R. \qquad\qquad (\text{Definition of } \boldsymbol{\beta}_\phi, 45e)$$

$\blacksquare$

**Theorem 4.3** (Counting in a Set)**.** *Under the conditions of Thm. 4.2, let $P = \{\boldsymbol{\pi}_k\}_{k=1}^K$ be a corpus of $K$ paths sampled i.i.d. from $\mathbf{\Pi}_{\mathcal{A}}(q_\iota)$, and let $\boldsymbol{Z} \stackrel{\text{def}}{=} \boldsymbol{\beta}_\phi(q_\iota) = \bigoplus_{\boldsymbol{\pi} \in \Pi(\mathcal{A}_\phi)} \boldsymbol{w}_\phi(\boldsymbol{\pi})$ be the sum of all path weights in $\mathcal{A}_\phi$. For any $n \in [R]_0$, we have*

$$(\boldsymbol{Z}^{\otimes K})_n = \begin{cases} \mathbb{P}(|P|_\phi = n) & \textbf{if } n < R \\ \mathbb{P}(|P|_\phi \geqslant R) & \textbf{if } n = R. \end{cases} \tag{20}$$

*Proof.* The case $K = 1$ is Thm. 4.2 at $q = q_\iota$: $\boldsymbol{Z} = \boldsymbol{\beta}_\phi(q_\iota)$ is the binning representation of $|\mathbf{\Pi}_{\mathcal{A}}(q_\iota)|_\phi$, i.e., $\boldsymbol{Z}_n = \mathbb{P}(|\mathbf{\Pi}_{\mathcal{A}}(q_\iota)|_\phi = n)$ for $n < R$ and $\boldsymbol{Z}_R = \mathbb{P}(|\mathbf{\Pi}_{\mathcal{A}}(q_\iota)|_\phi \geqslant R)$.

For $K > 1$, proceed by induction. The samples are i.i.d., so $|P|_\phi = |\boldsymbol{\pi}_K|_\phi + |\{\boldsymbol{\pi}_j\}_{j<K}|_\phi$ is a sum of two independent non-negative integer counts with binning representations $\boldsymbol{Z}$ (by Thm. 4.2) and $\boldsymbol{Z}^{\otimes K-1}$ (by the inductive hypothesis). Direct expansion of Eq. (11) shows that $\otimes$ is the overflow-accumulating convolution sending the pair of binning representations of independent $X, Y$ to the binning representation of $X + Y$: $\sum_{j=0}^n u_j v_{n-j} = \mathbb{P}(X + Y = n)$ for $n < R$, and $\sum_{\substack{0 \leqslant i,j \leqslant R \\ i+j \geqslant R}} u_i v_j = \mathbb{P}(X + Y \geqslant R)$ for $n = R$. Hence $\boldsymbol{Z}^{\otimes K} = \boldsymbol{Z}^{\otimes K-1} \otimes \boldsymbol{Z}$ is the binning representation of $|P|_\phi$. $\blacksquare$

**Theorem 4.4** (Sampling Path Event Counts)**.** *Under the conditions of Thm. 4.3 (a corpus $P$ of $K$ i.i.d. paths from $\mathbf{\Pi}_{\mathcal{A}}(q_\iota)$), let $N \in \mathbb{N}$ be a target total count and set the binning semiring order to $R = N + 1$. For any $k \in [K]$, $m \in [N]_0$, and $n \in [N - m]_0$,*

$$\mathbb{P}(|\boldsymbol{\pi}_k|_\phi = n \mid |P|_\phi = N, |\{\boldsymbol{\pi}_j\}_{j<k}|_\phi = m)$$
$$= \frac{(\boldsymbol{Z}^{\otimes K-k})_{N-m-n}}{(\boldsymbol{Z}^{\otimes K-k+1})_{N-m}} \boldsymbol{Z}_n. \tag{21}$$

*Proof.* We read the LHS of Eq. (21) as the probability that path $\boldsymbol{\pi}_k$ contributes exactly $n$ occurrences of $\phi$, conditional on the corpus $P$ totaling $N$ occurrences and the previously-sampled paths $\{\boldsymbol{\pi}_j\}_{j<k}$ contributing $m$ of them. By Bayes' rule,

$$\mathbb{P}(|\boldsymbol{\pi}_k|_\phi = n \mid |P|_\phi = N, |\{\boldsymbol{\pi}_j\}_{j<k}|_\phi = m) = \frac{\mathbb{P}(|\boldsymbol{\pi}_k|_\phi = n, |P|_\phi = N, |\{\boldsymbol{\pi}_j\}_{j<k}|_\phi = m)}{\mathbb{P}(|P|_\phi = N, |\{\boldsymbol{\pi}_j\}_{j<k}|_\phi = m)}. \tag{46}$$

The samples are i.i.d., so Thm. 4.3 applied to the disjoint subcorpora $\{\boldsymbol{\pi}_j\}_{j<k}$, $\{\boldsymbol{\pi}_k\}$, $\{\boldsymbol{\pi}_j\}_{j>k}$ gives binning representations $\boldsymbol{Z}^{\otimes k-1}$, $\boldsymbol{Z}$, $\boldsymbol{Z}^{\otimes K-k}$ for their independent $\phi$-counts. With $R = N + 1$, every coordinate accessed below lies in $[N]_0$, so we read exact (non-overflow) entries throughout.

For the numerator, $|P|_\phi = N$ together with the other two constraints forces $|\{\boldsymbol{\pi}_j\}_{j>k}|_\phi = N - m - n$:

$$\mathbb{P}(|\boldsymbol{\pi}_k|_\phi = n, |P|_\phi = N, |\{\boldsymbol{\pi}_j\}_{j<k}|_\phi = m)$$
$$= \mathbb{P}(|\boldsymbol{\pi}_k|_\phi = n, |\{\boldsymbol{\pi}_j\}_{j>k}|_\phi = N - m - n, |\{\boldsymbol{\pi}_j\}_{j<k}|_\phi = m) \tag{47a}$$
$$= \boldsymbol{Z}_n \cdot (\boldsymbol{Z}^{\otimes K-k})_{N-m-n} \cdot (\boldsymbol{Z}^{\otimes k-1})_m. \qquad\qquad (\text{Independence + Thm. 4.3, } 47b)$$

Similarly, $|P|_\phi = N$ with $|\{\boldsymbol{\pi}_j\}_{j<k}|_\phi = m$ is equivalent to $|\{\boldsymbol{\pi}_j\}_{j\geqslant k}|_\phi = N - m$:

$$\mathbb{P}(|P|_\phi = N, |\{\boldsymbol{\pi}_j\}_{j<k}|_\phi = m)$$

$$= \mathbb{P}(|\{\boldsymbol{\pi}_j\}_{j\geqslant k}|_\phi = N - m) \cdot \mathbb{P}(|\{\boldsymbol{\pi}_j\}_{j<k}|_\phi = m) \tag{48a}$$

$$= (\boldsymbol{Z}^{\otimes K-k+1})_{N-m} \cdot (\boldsymbol{Z}^{\otimes k-1})_m. \qquad \text{(Independence + Thm. 4.3, 48b)}$$

The $(\boldsymbol{Z}^{\otimes k-1})_m$ factor cancels in the ratio, yielding Eq. (21). ∎

We now consider sampling under the relaxed corpus-level constraint $|P|_\phi \geqslant N$ in place of $|P|_\phi = N$. To turn the exact-count binning representations of Thm. 4.3 into at-least probabilities, we left-multiply by the upper-triangular matrix of ones $\boldsymbol{M}$.

**Lemma E.1** (Tail Vector Interpretation). *Let $X$ be a non-negative integer random variable with binning representation $\boldsymbol{u} \in [\![\mathbb{K}]\!]_R$ of order $R$, meaning $u_i = \mathbb{P}(X = i)$ for $i < R$ and $u_R = \mathbb{P}(X \geqslant R)$. Let $\boldsymbol{M}$ be the upper-triangular matrix of ones. Then for any $n \in [R]_0$,*

$$(\boldsymbol{M}\boldsymbol{u})_n = \mathbb{P}(X \geqslant n). \tag{49}$$

*Proof.* $(\boldsymbol{M}\boldsymbol{u})_n = \sum_{j=n}^R u_j$ by definition of $\boldsymbol{M}$, and summing the disjoint events $\{X = n\}, \ldots, \{X = R - 1\}, \{X \geqslant R\}$ gives $\mathbb{P}(X \geqslant n)$. ∎

**Theorem E.1** (Sampling Path Event Counts (2)). *Under the setup of Thm. 4.4, define the tail vector $\boldsymbol{Z}' \overset{\text{def}}{=} \boldsymbol{M}\boldsymbol{Z}$, with $\boldsymbol{Z}'_n = \mathbb{P}(|\boldsymbol{\Pi}_{\mathcal{A}}(q_\iota)|_\phi \geqslant n)$ by Lemma E.1. For any $k \in [K]$, $m \in [N]_0$, and $n \in [N - m]_0$,*

$$\mathbb{P}(|\boldsymbol{\pi}_k|_\phi \geqslant n \mid |P|_\phi \geqslant N, |\{\boldsymbol{\pi}_j\}_{j<k}|_\phi = m) = \frac{\boldsymbol{Z}'_n (\boldsymbol{M}\boldsymbol{Z}^{\otimes K-k})_{N-m-n} + \sum_{j=n+1}^{N-m} \boldsymbol{Z}'_j (\boldsymbol{Z}^{\otimes K-k})_{N-m-j}}{(\boldsymbol{M}\boldsymbol{Z}^{\otimes K-k+1})_{N-m}}. \tag{50}$$

*Proof.* We read the LHS of Eq. (50) as the probability that path $\boldsymbol{\pi}_k$ contributes at least $n$ occurrences of $\phi$, conditional on the corpus $P$ having at least $N$ total and the previously-sampled paths $\{\boldsymbol{\pi}_j\}_{j<k}$ contributing exactly $m$. By Bayes' rule,

$$\mathbb{P}(|\boldsymbol{\pi}_k|_\phi \geqslant n \mid |P|_\phi \geqslant N, |\{\boldsymbol{\pi}_j\}_{j<k}|_\phi = m) = \frac{\mathbb{P}(|\boldsymbol{\pi}_k|_\phi \geqslant n, |P|_\phi \geqslant N, |\{\boldsymbol{\pi}_j\}_{j<k}|_\phi = m)}{\mathbb{P}(|P|_\phi \geqslant N, |\{\boldsymbol{\pi}_j\}_{j<k}|_\phi = m)}. \tag{51}$$

As in Thm. 4.4, Thm. 4.3 applied to the disjoint subcorpora $\{\boldsymbol{\pi}_j\}_{j<k}$, $\{\boldsymbol{\pi}_k\}$, $\{\boldsymbol{\pi}_j\}_{j>k}$ gives independent counts with exact-count binning representations $\boldsymbol{Z}^{\otimes k-1}$, $\boldsymbol{Z}$, $\boldsymbol{Z}^{\otimes K-k}$. Tail (at-least) probabilities are then $\boldsymbol{M}$-multiplied versions by Lemma E.1.

Let $X = |\boldsymbol{\pi}_k|_\phi$ and $Y = |\{\boldsymbol{\pi}_j\}_{j>k}|_\phi$. Path $\boldsymbol{\pi}_k$ faces two lower bounds on $X$: the LHS demand $n$, and the deficit $(N - m) - Y$ that $\boldsymbol{\pi}_k$ must cover for the corpus to reach $N$. So $X \geqslant \max(n, (N - m) - Y)$, and this max collapses into one of three sub-cases depending on $Y$: $Y \geqslant N - m$ (no deficit; $X \geqslant n$); $Y = N - m - j$ with $1 \leqslant j \leqslant n$ (deficit at most $n$, still $X \geqslant n$); or $Y = N - m - j$ with $j > n$ (deficit dominates, $X \geqslant j$). Independence then gives

$$\mathbb{P}(X \geqslant n, |P|_\phi \geqslant N, |\{\boldsymbol{\pi}_j\}_{j<k}|_\phi = m) = \boldsymbol{Z}'_n \cdot (\boldsymbol{M}\boldsymbol{Z}^{\otimes K-k})_{N-m} \cdot (\boldsymbol{Z}^{\otimes k-1})_m$$

$$+ \boldsymbol{Z}'_n \sum_{j=1}^n (\boldsymbol{Z}^{\otimes K-k})_{N-m-j} \cdot (\boldsymbol{Z}^{\otimes k-1})_m$$

$$+ \sum_{j=n+1}^{N-m} \boldsymbol{Z}'_j \cdot (\boldsymbol{Z}^{\otimes K-k})_{N-m-j} \cdot (\boldsymbol{Z}^{\otimes k-1})_m \tag{52a}$$

$$= \left( \boldsymbol{Z}'_n \cdot (\boldsymbol{M}\boldsymbol{Z}^{\otimes K-k})_{N-m-n} + \sum_{j=n+1}^{N-m} \boldsymbol{Z}'_j \cdot (\boldsymbol{Z}^{\otimes K-k})_{N-m-j} \right) (\boldsymbol{Z}^{\otimes k-1})_m, \tag{52b}$$

using $(\boldsymbol{M}\boldsymbol{Z}^{\otimes K-k})_{N-m} + \sum_{j=1}^n (\boldsymbol{Z}^{\otimes K-k})_{N-m-j} = (\boldsymbol{M}\boldsymbol{Z}^{\otimes K-k})_{N-m-n}$ to combine the first two summands (extending the upper-tail sum down by $n$ exact terms).

The denominator is the joint probability that the suffix totals at least $N - m$ and the prefix exactly $m$, independent:

$$\mathbb{P}(|P|_\phi \geqslant N, |\{\boldsymbol{\pi}_j\}_{j<k}|_\phi = m) = \mathbb{P}(|\{\boldsymbol{\pi}_j\}_{j \geqslant k}|_\phi \geqslant N - m) \cdot \mathbb{P}(|\{\boldsymbol{\pi}_j\}_{j<k}|_\phi = m) \tag{53a}$$

$$= (\boldsymbol{M}\boldsymbol{Z}^{\otimes K-k+1})_{N-m} \cdot (\boldsymbol{Z}^{\otimes k-1})_m. \tag{53b}$$

The $(\boldsymbol{Z}^{\otimes k-1})_m$ factor cancels in the ratio, yielding Eq. (50). ∎

We now consider the at-least-once intervention used in §5: the corpus-level constraint is on the *number of strings* containing at least one occurrence of $\phi$ rather than on the total count. Replacing the raw count by the per-path indicator $\mathbb{1}\{\phi\}$ collapses each path to a Bernoulli draw.

**Theorem E.2** (Sampling Path Event Counts (3)). *Under the setup of Thm. 4.3 with $R = 1$ and corpus size $K$, $\boldsymbol{Z}_1 = \mathbb{P}(|\boldsymbol{\Pi}_\mathcal{A}(q_\iota)|_\phi \geqslant 1)$ is the per-path probability of containing at least one occurrence (Thm. 4.2). Define the per-path indicator $\mathbb{1}\{\phi\}(\boldsymbol{\pi}) \overset{\text{def}}{=} \mathbb{1}\{|\boldsymbol{\pi}|_\phi \geqslant 1\}$ and its corpus aggregation $|P|_{\mathbb{1}\{\phi\}} \overset{\text{def}}{=} \sum_{j=1}^{K} \mathbb{1}\{\phi\}(\boldsymbol{\pi}_j)$, and let $M \in [K]_0$ be a target value of $|P|_{\mathbb{1}\{\phi\}}$. For $k \in [K]$ and $m \in [\min(M, k-1)]_0$ with $M - m \leqslant K - k + 1$,*

$$\mathbb{P}(|\boldsymbol{\pi}_k|_\phi \geqslant 1 \mid |P|_{\mathbb{1}\{\phi\}} = M, |\{\boldsymbol{\pi}_j\}_{j<k}|_{\mathbb{1}\{\phi\}} = m) = \frac{M - m}{K - k + 1}. \tag{54}$$

*In words: given that $m$ of the first $k - 1$ paths in the corpus contain at least one occurrence of $\phi$ (out of $M$ such paths required across all $K$), path $\boldsymbol{\pi}_k$ contains at least one with probability $(M - m)/(K - k + 1)$—the remaining at-least-one paths divided by the remaining paths to sample.*

*Proof.* Let $X_j \overset{\text{def}}{=} \mathbb{1}\{\phi\}(\boldsymbol{\pi}_j) \in \{0, 1\}$. Each $X_j$ indicates a single event of probability $\boldsymbol{Z}_1$ and is therefore Bernoulli($\boldsymbol{Z}_1$); the $X_j$ are i.i.d. since the paths are. The conditioning event $\{|P|_{\mathbb{1}\{\phi\}} = M, |\{\boldsymbol{\pi}_j\}_{j<k}|_{\mathbb{1}\{\phi\}} = m\}$ rewrites as $\{\sum_{j=1}^{K} X_j = M, \sum_{j<k} X_j = m\}$, equivalently $\{\sum_{j<k} X_j = m, \sum_{j \geqslant k} X_j = M - m\}$. Since the prefix sum is independent of $\{X_j\}_{j \geqslant k}$, conditioning on the suffix sum alone gives the same conditional. Then, by Bayes' rule and the binomial pmf,

$$\mathbb{P}(|\boldsymbol{\pi}_k|_\phi \geqslant 1 \mid |P|_{\mathbb{1}\{\phi\}} = M, |\{\boldsymbol{\pi}_j\}_{j<k}|_{\mathbb{1}\{\phi\}} = m)$$

$$= \mathbb{P}\left(X_k = 1 \mid \textstyle\sum_{j \geqslant k} X_j = M - m\right) \quad\quad \text{(prefix independent of } \{X_j\}_{j \geqslant k}, 55)$$

$$= \frac{\mathbb{P}\left(X_k = 1, \sum_{j>k} X_j = M - m - 1\right)}{\mathbb{P}\left(\sum_{j \geqslant k} X_j = M - m\right)} \quad\quad \text{(Bayes; substitute } X_k = 1, 56)$$

$$= \frac{\mathbb{P}(X_k = 1)\,\mathbb{P}\left(\sum_{j>k} X_j = M - m - 1\right)}{\mathbb{P}\left(\sum_{j \geqslant k} X_j = M - m\right)} \quad\quad \text{(independence of } X_k \text{ and } \{X_j\}_{j>k}, 57)$$

$$= \frac{\boldsymbol{Z}_1\binom{K-k}{M-m-1}\boldsymbol{Z}_1^{M-m-1}(1-\boldsymbol{Z}_1)^{K-k-M+m+1}}{\binom{K-k+1}{M-m}\boldsymbol{Z}_1^{M-m}(1-\boldsymbol{Z}_1)^{K-k+1-M+m}} \quad\quad \text{(Bernoulli + binomial pmf, 58)}$$

$$= \frac{\binom{K-k}{M-m-1}}{\binom{K-k+1}{M-m}} \quad\quad \text{(cancel } \boldsymbol{Z}_1^{M-m} \text{ and } (1-\boldsymbol{Z}_1)^{K-k+1-M+m}, 59)$$

$$= \frac{M - m}{K - k + 1}. \quad\quad \left(\binom{n}{r-1}/\binom{n+1}{r} = r/(n+1) \text{ with } n = K - k, r = M - m, 60\right)$$

∎

The corresponding per-path sampling step does not need a new corollary: once $n_k \in \{0, 1\}$ has been drawn for path $\boldsymbol{\pi}_k$, we sample $\boldsymbol{\pi}_k$ via Thm. 4.5 with $n = 0$ (a path with no occurrences) or Thm. E.3 with $n = 1$ (a path with at least one).

**Theorem 4.5** (Constrained Sampling). *Under the conditions of Thm. 4.4, fix $n \in [N]_0$ and define a lifted PFA $\mathcal{A}'$ over the state set $Q \times [n]_0$ (see App. E). Let $\boldsymbol{\Pi}_{\mathcal{A}'}(q_\iota)$ be a random path drawn from $\mathcal{A}'$. For any $\boldsymbol{\pi} \in \Pi(\mathcal{A}')$,*

$$\mathbb{P}(\boldsymbol{\Pi}_\mathcal{A}(q_\iota) = \boldsymbol{\pi}_\mathcal{A} \mid |\boldsymbol{\Pi}_\mathcal{A}(q_\iota)|_\phi = n) = \mathbb{P}(\boldsymbol{\Pi}_{\mathcal{A}'}(q_\iota) = \boldsymbol{\pi}).$$

*Proof.* The lifted PFA $\mathcal{A}' = (Q', \Sigma, \delta', \lambda', \rho')$ is defined as follows. Write $v = \phi_\delta(q \xrightarrow{\sigma/w} q')$ for the event count of a transition; let $Z(q) \stackrel{\text{def}}{=} \beta_\phi(q)$. Then:

(i) $Q' = Q \times [n]_0$,

(ii) $\delta' : ((q,r), \sigma, (q', r-v)) \mapsto \frac{1}{Z(q)_r} \left( \mathcal{L}_{\phi_\delta}(q \xrightarrow{\sigma/w} q') \otimes \beta_\phi(q') \right)_r$,

(iii) $\lambda' : (q,r) \mapsto \mathbb{1}\{q = q_\iota, r = n\}$,

(iv) $\rho' : (q,r) \mapsto \frac{1}{Z(q)_r} \mathcal{L}_\rho(q)_r$.

The state coordinate $r$ tracks the remaining target count: each transition decrements it by $v$, and the final weight fires only when $r = \phi_\rho(q)$.

We first show that the weights of $\delta'$ and $\rho'$ sum to 1 for any $q \in Q$ and $r \in [n]_0$. Note that by definition, we have the following equalities

$$\delta'((q,r), \sigma, (q', r-v)) \stackrel{\text{def}}{=} \frac{1}{\beta_\phi(q)_r} \cdot \left( \mathcal{L}_{\phi_\delta}(q \xrightarrow{\sigma/w} q') \otimes \beta_\phi(q') \right)_r \tag{61a}$$

$$= \frac{1}{\beta_\phi(q)_r} \cdot w \cdot \left( \beta_\phi(q') \right)_{r - \phi_\delta(q \xrightarrow{\sigma/w} q')} \tag{61b}$$

$$\rho'(q,r) \stackrel{\text{def}}{=} \frac{1}{\beta_\phi(q)_r} \cdot \mathcal{L}_\rho(q)_r \tag{61c}$$

$$= \frac{1}{\beta_\phi(q)_r} \cdot \mathbb{1}\{r = \phi_\rho(q)\} \rho(q) \tag{61d}$$

From the equality

$$\beta_{\mathcal{A}}(q) = \bigoplus_{q \xrightarrow{\sigma/w} q'} w \otimes \beta_{\mathcal{A}}(q') \oplus \rho(q) \tag{62}$$

in a general WFA, we have that[7]

$$\left( \beta_\phi(q) \right)_r = \bigoplus_{q \xrightarrow{\sigma/w} q'} \left( w \otimes \beta_\phi(q') \right)_r \oplus \left( \rho_{\mathcal{A}_\phi}(q) \right)_r \tag{63}$$

$$= \bigoplus_{q \xrightarrow{\sigma/w} q'} \left( \mathcal{L}_{\phi_\delta}(q \xrightarrow{\sigma/w} q') \otimes \beta_\phi(q') \right)_r \oplus \left( \rho_{\mathcal{A}_\phi}(q) \right)_r \tag{64}$$

$$= \bigoplus_{q \xrightarrow{\sigma/w} q'} w \cdot \beta_\phi(q')_{r - \phi_\delta(q \xrightarrow{\sigma/w} q')} \oplus \left( \rho_{\mathcal{A}_\phi}(q) \right)_r \tag{65}$$

$$= \bigoplus_{q \xrightarrow{\sigma/w} q'} w \cdot \beta_\phi(q')_{r - \phi_\delta(q \xrightarrow{\sigma/w} q')} \oplus \mathbb{1}\{r = \phi_\rho(q)\} \rho(q) \tag{66}$$

Dividing by $\beta_\phi(q)_r$ matches the unfolded $\delta'$ and $\rho'$ above, so $\sum_{q \xrightarrow{\sigma/w} q' \in \delta} \delta'((q,r), \sigma, (q', r-v)) + \rho'(q,r) = 1$ for every $(q,r)$—hence $\mathcal{A}'$ is a valid PFA, and the path-probability formula below applies.

For $\pi = (q_\iota, n) \xrightarrow{a_1/w_1} \cdots \xrightarrow{a_m/w_m} (q_m, r_m)$—where $w_t$ denotes the corresponding base-automaton transition weight—we then have, using the unfolded $\delta'$ and $\rho'$ above (throughout this proof, write $|\pi_{\leqslant t}|_\phi \stackrel{\text{def}}{=} \sum_{s=1}^{t} \phi_\delta(q_{s-1} \xrightarrow{a_s/w_s} q_s)$ for the transition-only feature sum on the first $t$ transitions of $\pi$, with $|\pi_{<t}|_\phi \stackrel{\text{def}}{=} |\pi_{\leqslant t-1}|_\phi$; these omit the init and final contributions

---

[7]For conciseness, we assume $v_m = 0$ for $m < 0$ and $m > R$.

to $|\boldsymbol{\pi}|_\phi$):

$$\mathbb{P}(\boldsymbol{\Pi}_{\mathcal{A}'}(q_\iota) = \boldsymbol{\pi}) = \prod_{t=1}^{m} \delta'((q_{t-1}, r_{t-1}), a_t, (q_t, r_t)) \cdot \rho'(q_m, r_m) \tag{67a}$$

$$= \prod_{t=1}^{m} \frac{1}{\left(\boldsymbol{\beta}_\phi(q_{t-1})\right)_{n-|\boldsymbol{\pi}_{<t}|_\phi}} \cdot \left(\mathcal{L}_{\phi_\delta}(q_{t-1} \xrightarrow{a_t/w_t} q_t) \otimes \boldsymbol{\beta}_\phi(q_t)\right)_{n-|\boldsymbol{\pi}_{<t}|_\phi}$$
$$\cdot \frac{1}{\left(\boldsymbol{\beta}_\phi(q_m)\right)_{n-|\boldsymbol{\pi}_{\leqslant m}|_\phi}} \cdot \left(\mathcal{L}_\rho(q_m)\right)_{n-|\boldsymbol{\pi}_{\leqslant m}|_\phi} \qquad \text{(Eqs. (61a) and (61c), 67b)}$$

$$= \prod_{t=1}^{m} \frac{1}{\left(\boldsymbol{\beta}_\phi(q_{t-1})\right)_{n-|\boldsymbol{\pi}_{<t}|_\phi}} \cdot w_t \cdot \left(\boldsymbol{\beta}_\phi(q_t)\right)_{n-|\boldsymbol{\pi}_{<t}|_\phi - \phi_\delta(q_{t-1} \xrightarrow{a_t/w_t} q_t)}$$
$$\cdot \frac{1}{\left(\boldsymbol{\beta}_\phi(q_m)\right)_{n-|\boldsymbol{\pi}_{\leqslant m}|_\phi}} \cdot \left(\mathcal{L}_\rho(q_m)\right)_{n-|\boldsymbol{\pi}_{\leqslant m}|_\phi} \qquad \text{(Eq. (61b), 67c)}$$

$$= \prod_{t=1}^{m} \frac{1}{\left(\boldsymbol{\beta}_\phi(q_{t-1})\right)_{n-|\boldsymbol{\pi}_{<t}|_\phi}} \cdot w_t \cdot \left(\boldsymbol{\beta}_\phi(q_t)\right)_{n-|\boldsymbol{\pi}_{\leqslant t}|_\phi}$$
$$\cdot \frac{1}{\left(\boldsymbol{\beta}_\phi(q_m)\right)_{n-|\boldsymbol{\pi}_{\leqslant m}|_\phi}} \cdot \left(\mathcal{L}_\rho(q_m)\right)_{n-|\boldsymbol{\pi}_{\leqslant m}|_\phi} \qquad (|\boldsymbol{\pi}_{<t}|_\phi + \phi_\delta(q_{t-1} \xrightarrow{a_t/w_t} q_t) = |\boldsymbol{\pi}_{\leqslant t}|_\phi, \text{67d})$$

$$= \frac{1}{\left(\boldsymbol{\beta}_\phi(q_\iota)\right)_n} \cdot \prod_{t=1}^{m} w_t \cdot \left(\mathcal{L}_\rho(q_m)\right)_{n-|\boldsymbol{\pi}_{\leqslant m}|_\phi} \qquad \text{(Telescoping product, 67e)}$$

$$= \frac{1}{\left(\boldsymbol{\beta}_\phi(q_\iota)\right)_n} \cdot \prod_{t=1}^{m} w_t \cdot \mathbb{1}\{n - |\boldsymbol{\pi}_{\leqslant m}|_\phi = \phi_\rho(q_m)\} \cdot \rho(q_m) \qquad \text{(definition of } \mathcal{L}_\rho \text{ (Definition 4.1), 67f)}$$

$$= \frac{1}{\left(\boldsymbol{\beta}_\phi(q_\iota)\right)_n} \cdot \prod_{t=1}^{m} w_t \cdot \rho(q_m) \cdot \mathbb{1}\{n - |\boldsymbol{\pi}_{\leqslant m}|_\phi = \phi_\rho(q_m)\} \tag{67g}$$

$$= \frac{1}{\left(\boldsymbol{\beta}_\phi(q_\iota)\right)_n} \cdot \mathbb{P}(\boldsymbol{\Pi}_{\mathcal{A}}(q_\iota) = \boldsymbol{\pi}_{\mathcal{A}}) \cdot \mathbb{1}\{n - |\boldsymbol{\pi}_{\leqslant m}|_\phi = \phi_\rho(q_m)\} \qquad \text{(path probability in } \mathcal{A} \text{ (Eq. (18)), 67h)}$$

$$= \frac{1}{\left(\boldsymbol{\beta}_\phi(q_\iota)\right)_n} \cdot \mathbb{P}(\boldsymbol{\Pi}_{\mathcal{A}}(q_\iota) = \boldsymbol{\pi}_{\mathcal{A}}, |\boldsymbol{\pi}|_\phi = n) \qquad (\phi_\lambda(q_\iota) = 0 \text{ implies } |\boldsymbol{\pi}|_\phi = |\boldsymbol{\pi}_{\leqslant m}|_\phi + \phi_\rho(q_m), \text{67i)}$$

$$= \frac{1}{\mathbb{P}(|\boldsymbol{\Pi}_{\mathcal{A}}(q_\iota)|_\phi = n)} \cdot \mathbb{P}(\boldsymbol{\Pi}_{\mathcal{A}}(q_\iota) = \boldsymbol{\pi}_{\mathcal{A}}, |\boldsymbol{\pi}|_\phi = n) \qquad \text{(Thm. 4.2, 67j)}$$

$$= \mathbb{P}(\boldsymbol{\Pi}_{\mathcal{A}}(q_\iota) = \boldsymbol{\pi}_{\mathcal{A}} \mid |\boldsymbol{\Pi}_{\mathcal{A}}(q_\iota)|_\phi = n). \tag{67k}$$

∎

For the at-least variant of Thm. 4.5—where the per-path constraint is $|\boldsymbol{\Pi}_{\mathcal{A}}(q_\iota)|_\phi \geqslant n$ instead of $= n$—we construct an analogous lifted PFA $\mathcal{A}''$ using the accumulated-tail backward weights $\boldsymbol{M}\boldsymbol{\beta}_\phi$ (Lemma E.1) in place of the exact-count $\boldsymbol{\beta}_\phi$.

**Theorem E.3** (Constrained String Sampling (2)). *Define $\mathcal{A}''$ analogously to $\mathcal{A}'$ in Thm. 4.5, but using the accumulated-tail backward weights $\boldsymbol{Z}'(q) = \boldsymbol{M}\boldsymbol{\beta}_\phi(q)$ from Lemma E.1 in place of $\boldsymbol{\beta}_\phi(q)$, with the counter clamped at $0$ on decrement. Let $\boldsymbol{\Pi}_{\mathcal{A}''}(q_\iota)$ be a random path drawn from $\mathcal{A}''$. For any per-path target $n \in [N]_0$ and any $\boldsymbol{\pi} \in \Pi(\mathcal{A}'')$,*

$$\mathbb{P}(\boldsymbol{\Pi}_{\mathcal{A}}(q_\iota) = \boldsymbol{\pi}_{\mathcal{A}} \mid |\boldsymbol{\Pi}_{\mathcal{A}}(q_\iota)|_\phi \geqslant n) = \mathbb{P}(\boldsymbol{\Pi}_{\mathcal{A}''}(q_\iota) = \boldsymbol{\pi}). \tag{68}$$

*Proof.* By definition, we have the following equalities

$$\delta''((q, r), \sigma, (q', \max(0, r - v))) \stackrel{\text{def}}{=} \frac{1}{\boldsymbol{Z}'(q)_r} \cdot \left(\boldsymbol{M}(\mathcal{L}_{\phi_\delta}(q \xrightarrow{\sigma/w} q') \otimes \boldsymbol{\beta}_\phi(q'))\right)_r \tag{69a}$$

$$= \frac{1}{\boldsymbol{Z}'(q)_r} \cdot w \cdot \boldsymbol{Z}'(q')_{\max(0, r - \phi_\delta(q \xrightarrow{\sigma/w} q'))} \tag{69b}$$

$$\rho''(q, r) \stackrel{\text{def}}{=} \frac{1}{\boldsymbol{Z}'(q)_r} \cdot \left(\boldsymbol{M}\mathcal{L}_\rho(q)\right)_r \tag{69c}$$

$$= \frac{1}{\boldsymbol{Z}'(q)_r} \cdot \mathbb{1}\{r \leqslant \phi_\rho(q)\}\rho(q). \tag{69d}$$

The next-state probabilities sum to 1 at every $(q, r)$ by the same argument as in Thm. 4.5 (applied to $\boldsymbol{Z}'$ in place of $\boldsymbol{Z}$). The path-probability derivation mirrors Thm. 4.5: for $\boldsymbol{\pi} = (q_\iota, n) \xrightarrow{a_1/w_1} \cdots \xrightarrow{a_m/w_m} (q_m, r_m)$ (with $w_t$ the corresponding base-automaton transition weight), the same telescoping product gives the analog of Thm. 4.5's derivation up to the final-state factor: there the indicator $\mathbb{1}\{r_m = \phi_\rho(q_m)\}$ captured the exact-count event, whereas here $\rho''$ contributes $\mathbb{1}\{r_m \leqslant \phi_\rho(q_m)\}$, which equals $\mathbb{1}\{|\boldsymbol{\pi}|_\phi \geqslant n\}$ in both clamp regimes (unclamped: $r_m = n - |\boldsymbol{\pi}_{\leqslant m}|_\phi$, so the indicator reads $|\boldsymbol{\pi}_{\leqslant m}|_\phi + \phi_\rho(q_m) \geqslant n$, which equals $|\boldsymbol{\pi}|_\phi \geqslant n$ via $\phi_\lambda(q_\iota) = 0$; clamped at $r_m = 0$: the indicator is vacuously true and $|\boldsymbol{\pi}_{\leqslant m}|_\phi \geqslant n$ already, so $|\boldsymbol{\pi}|_\phi \geqslant n$ as well). Substituting $\boldsymbol{Z}'$ for $\boldsymbol{\beta}_\phi$ and $\mathbb{1}\{|\boldsymbol{\pi}|_\phi \geqslant n\}$ for the exact-count indicator throughout the Thm. 4.5 chain yields

$$\mathbb{P}(\boldsymbol{\Pi}_{\mathcal{A}''}(q_\iota) = \boldsymbol{\pi}) = \mathbb{P}(\boldsymbol{\Pi}_{\mathcal{A}}(q_\iota) = \boldsymbol{\pi}_{\mathcal{A}} \mid |\boldsymbol{\Pi}_{\mathcal{A}}(q_\iota)|_\phi \geqslant n). \tag{70}$$

∎

## F. Naïve Sampling and Runtime

### F.1. Naïve Sampling Baselines

**Rejection Sampling.** Rejection sampling is the most direct approach to sample from $\mathbb{P}(\mathrm{D} \mid \mathrm{A} = \mathcal{A}, \mathrm{do}(\mathrm{D} \in \mathcal{P}))$. Suppose we target transitions emitting a and wish to sample a corpus of $K$ strings in which a occurs exactly $N$ times overall. Draw a corpus from $p$ unconstrained; if the total a-count is $N$, return it; otherwise discard and repeat. The acceptance probability decays exponentially in the gap between $N$ and the expected a-count of an unconstrained corpus, so the expected number of corpus draws grows exponentially in that gap.

**Intersection-based Sampling.** A more structured approach decomposes the problem into per-string sampling. First, fix the corpus size $K$ and sample per-string occurrence counts $n_1, \ldots, n_K \in \mathbb{N}$ with $\sum_{k=1}^K n_k = N$. This reduces the task to drawing, for each per-string count $n_k$, a string $\boldsymbol{\sigma} \sim p(\cdot \mid \boldsymbol{\sigma}$ has exactly $n_k$ symbols a). Using standard automata-theoretic constructions, we can sample each such string by: *(i)* constructing an FA $\mathcal{A}_{a,n_k}$ that recognizes strings with exactly $n_k$ a's; *(ii)* intersecting $\mathcal{A}_{a,n_k}$ with $\mathcal{A}$ to obtain $\mathcal{A}_{a,n_k} \cap \mathcal{A}$, which recognizes strings with exactly $n_k$ a's that are also accepted by $\mathcal{A}$; *(iii)* renormalizing $\mathcal{A}_{a,n_k} \cap \mathcal{A}$ via weight pushing (Mohri, 2009) to obtain a PFA; and *(iv)* ancestral-sampling a string from this PFA. Intersection yields an automaton with $\mathcal{O}(|Q|n_k)$ states, and weight pushing is cubic in the number of states, so the total cost across $n_k \in \{0, \ldots, N\}$ is $\mathcal{O}(|Q|^3 N^4)$. The constrained sampler of §4.3 improves this to $\mathcal{O}((|Q|^3 + K)N^2 + K\ell)$ (App. F.2).

### F.2. Runtime of Constrained Sampling

We analyze the runtime of `Preprocess`, `SamplePath`, and `SampleCorpus` from §4.3 in turn. Let $|\delta|$ be the number of transitions of $\mathcal{A}$, $K$ the number of strings, $N$ the corpus-level target count, and $\ell$ a bound on the length of the sampled paths.

`Preprocess`. Lifting $\mathcal{A}$ to the binning automaton $\mathcal{A}_\phi$ touches each transition once: $\mathcal{O}(|\delta|)$. Computing the backward weights $\boldsymbol{\beta}_\phi$ with Lehmann's algorithm (Lehmann, 1977) performs $\mathcal{O}(|Q|^3)$ binning-semiring operations, each costing $C = \mathcal{O}(N^2)$ in a general base semiring or $\mathcal{O}(N \log N)$ using FFT over the reals, for $\mathcal{O}(|Q|^3 N^2)$ total. Tabulating the per-state outgoing transitions $\mathcal{E}$ is another $\mathcal{O}(|\delta|)$ pass. Precomputing the per-action binning-weight table $\boldsymbol{W}$ costs $\mathcal{O}((|\delta| + |Q|)N)$: each lifted transition weight $\delta_\phi(q, a, q')$ has a single nonzero coordinate, so $\delta_\phi(q, a, q') \otimes \boldsymbol{\beta}_\phi(q')$

reduces to a scaled shift of $\boldsymbol{\beta}_\phi(q')$ computable in $\mathcal{O}(N)$ rather than a full $\mathcal{O}(N^2)$ convolution (and $\rho_\phi(q)$ supplies the EOS slot directly). This is dominated by the cubic Lehmann pass. Adding these, `Preprocess` runs in $\mathcal{O}(|Q|^3 N^2)$ time, or $\mathcal{O}(|Q|^3 N \log N)$ with FFT.

`SamplePath` **(one call).** At each step, `SamplePath` draws one action from the distribution over $\mathcal{E}(q) \cup \{\text{EOS}\}$ proportional to the slice $\boldsymbol{W}(q, \cdot)_{n-j}$. Drawing from the slice naively costs $\mathcal{O}(\max_q |\mathcal{E}(q)|) \leqslant \mathcal{O}(|\Sigma|)$ per step. Precomputing an alias table (Walker, 1977; Vose, 1991) for each state $q$ and remaining count $r \in [N]_0$ (the index $n-j$ at which $\boldsymbol{W}(q, \cdot)$ is read) instead makes every draw $\mathcal{O}(1)$; building these tables from $\boldsymbol{W}$ costs $\mathcal{O}((|\delta| + |Q|)N)$ (one pass over each outgoing transition and the EOS slot at each $r$), dominated by the cubic Lehmann pass. A path of length $\ell$ then costs $\mathcal{O}(\ell)$.

`SampleCorpus`. `SampleCorpus` first invokes `Preprocess` at the cost above. Phase 1 computes $\boldsymbol{Z}^{\otimes k}$ for $k \in \{1, \dots, K-1\}$ iteratively (one binning multiplication per $k$) and draws each $n_k$ by scanning $\mathcal{O}(N)$ proportional weights, for $\mathcal{O}(KN^2)$ (or $\mathcal{O}(KN \log N)$ with FFT). Phase 2 makes $K$ calls to `SamplePath`, each $\mathcal{O}(\ell)$, for $\mathcal{O}(K\ell)$. Adding the three contributions gives

$$\mathcal{O}\big((|Q|^3 + K)N^2 + K\ell\big),$$

or, with FFT in the binning multiplication,

$$\mathcal{O}\big((|Q|^3 + K)N \log N + K\ell\big).$$

Whether the cubic term $(|Q|^3 N^2)$, the per-string-count term $(KN^2)$, or the path-sampling term $(K\ell)$ dominates depends on the relative sizes of $|Q|$, $K$, $N$, and $\ell$. In practice, we find this approach to be two orders of magnitude faster than rejection sampling.

## G. Targeted KL Divergence via Decomposition

We are interested in measuring the learnability of targeted features. To this end, we derive a decomposed KL divergence that works on the transition, symbol, or state level between an automaton and a trained LM.

Let $p_{\mathcal{A}}$ be an LM defined by a DPFA $\mathcal{A}$ over states $Q$ and symbols $\Sigma$. Any string $\boldsymbol{\sigma}$ sampled from $p_{\mathcal{A}}$ decomposes into transitions consisting of states $q$, symbols $\sigma$, and weights $w$. Given another LM $p_{\boldsymbol{\theta}}$ and an event function $\phi$, we decompose the KL divergence to analyze how well $p_{\boldsymbol{\theta}}$ captures the target transitions $\mathcal{T}$ (§4). At each step, $p_{\mathcal{A}}$ takes transition $\delta$ with probability $w$, while $p_{\boldsymbol{\theta}}$ predicts the next symbol given the history $\boldsymbol{\sigma}_{<t}$ of all symbols preceding position $t$. As in §2, the next-symbol distribution $p_{\mathcal{A}}(\cdot \mid q)$ is supported on $\Sigma_{\text{EOS}}$ with $p_{\mathcal{A}}(\text{EOS} \mid q) = \rho(q)$. Recall the prefix probability $\overrightarrow{p}$ from Eq. (1).

Throughout, let

$$p_\pi(q) \stackrel{\text{def}}{=} \sum_{\boldsymbol{\sigma} \in \Sigma^*} \overrightarrow{p}(\boldsymbol{\sigma}) \mathbb{1}\{\hat{\delta}(\boldsymbol{\sigma}) = q\}, \qquad p_\pi(q, \sigma) \stackrel{\text{def}}{=} p_\pi(q) p_{\mathcal{A}}(\sigma \mid q), \qquad p_\pi(\sigma) \stackrel{\text{def}}{=} \sum_{q \in Q} p_\pi(q, \sigma). \tag{71}$$

We have that

$$\mathrm{D}_{\mathrm{KL}}(p_{\mathcal{A}} \parallel p_{\boldsymbol{\theta}}) = \mathbb{E}_{\boldsymbol{\sigma} \sim p_{\mathcal{A}}}\Big[\log \frac{p_{\mathcal{A}}(\boldsymbol{\sigma})}{p_{\boldsymbol{\theta}}(\boldsymbol{\sigma})}\Big] \tag{72a}$$

$$= \sum_{\boldsymbol{\sigma} \in \Sigma^*} \overrightarrow{p}(\boldsymbol{\sigma}) \mathrm{D}_{\mathrm{KL}}\big(p_{\mathcal{A}}(\cdot \mid \boldsymbol{\sigma}) \parallel p_{\boldsymbol{\theta}}(\cdot \mid \boldsymbol{\sigma})\big) \qquad \text{(chain rule on prefixes, 72b)}$$

**State-wise decomposition.** Group every history by the state it reaches, $q = \hat{\delta}(\boldsymbol{\sigma})$:

$$\mathrm{D}_{\mathrm{KL}}(p_{\mathcal{A}} \parallel p_{\boldsymbol{\theta}}) = \sum_{q \in Q} \sum_{\substack{\boldsymbol{\sigma} \in \Sigma^* \\ \hat{\delta}(\boldsymbol{\sigma}) = q}} \overrightarrow{p}(\boldsymbol{\sigma}) \mathrm{D}_{\mathrm{KL}}\big(p_{\mathcal{A}}(\cdot \mid q) \| p_{\boldsymbol{\theta}}(\cdot \mid \boldsymbol{\sigma})\big) \qquad \text{(insert } \mathbb{1}\{\hat{\delta}(\boldsymbol{\sigma}) = q\}, \text{73a)}$$

$$= \sum_{q \in Q} p_\pi(q) \underbrace{\sum_{\substack{\boldsymbol{\sigma} \in \Sigma^* \\ \hat{\delta}(\boldsymbol{\sigma}) = q}} \frac{\overrightarrow{p}(\boldsymbol{\sigma})}{p_\pi(q)} \mathrm{D}_{\mathrm{KL}}\big(p_{\mathcal{A}}(\cdot \mid q) \parallel p_{\boldsymbol{\theta}}(\cdot \mid \boldsymbol{\sigma})\big)}_{\text{Unweighted per-state contribution}} \tag{73b}$$

**Transition-wise decomposition.** Insert the next symbol $\sigma$ and write $p_\pi(q, \sigma) = p_\pi(q) p_\mathcal{A}(\sigma \mid q)$:

$$D_{\mathrm{KL}}(p_\mathcal{A} \parallel p_\theta) = \sum_{q \in Q} p_\pi(q) \sum_{\sigma \in \Sigma_{\mathrm{EOS}}} p_\mathcal{A}(\sigma \mid q) \sum_{\substack{\boldsymbol{\sigma} \in \Sigma^* \\ \widehat{\delta}(\boldsymbol{\sigma}) = q}} \frac{\overrightarrow{p}(\boldsymbol{\sigma})}{p_\pi(q)} \log \frac{p_\mathcal{A}(\sigma \mid q)}{p_\theta(\sigma \mid \boldsymbol{\sigma})} \tag{74a}$$

$$= \sum_{(q, \sigma) \in Q \times \Sigma_{\mathrm{EOS}}} p_\pi(q, \sigma) \underbrace{\sum_{\substack{\boldsymbol{\sigma} \in \Sigma^* \\ \widehat{\delta}(\boldsymbol{\sigma}) = q}} \frac{\overrightarrow{p}(\boldsymbol{\sigma})}{p_\pi(q)} \log \frac{p_\mathcal{A}(\sigma \mid q)}{p_\theta(\sigma \mid \boldsymbol{\sigma})}}_{\text{Unweighted per-transition contribution}} \tag{74b}$$

$$\tag{74c}$$

**Symbol-wise decomposition.** Marginalize over states to obtain the symbol prior $p_\pi(\sigma) = \sum_{q \in Q} p_\pi(q, \sigma)$:

$$D_{\mathrm{KL}}(p_\mathcal{A} \parallel p_\theta) = \sum_{\sigma \in \Sigma_{\mathrm{EOS}}} \sum_{\boldsymbol{\sigma} \in \Sigma^*} \overrightarrow{p}(\boldsymbol{\sigma}) \, p_\mathcal{A}(\sigma \mid \widehat{\delta}(\boldsymbol{\sigma})) \log \frac{p_\mathcal{A}(\sigma \mid \widehat{\delta}(\boldsymbol{\sigma}))}{p_\theta(\sigma \mid \boldsymbol{\sigma})} \tag{75a}$$

$$= \sum_{\sigma \in \Sigma_{\mathrm{EOS}}} p_\pi(\sigma) \underbrace{\sum_{\boldsymbol{\sigma} \in \Sigma^*} \frac{\overrightarrow{p}(\boldsymbol{\sigma}) p_\mathcal{A}(\sigma \mid \widehat{\delta}(\boldsymbol{\sigma}))}{p_\pi(\sigma)} \log \frac{p_\mathcal{A}(\sigma \mid \widehat{\delta}(\boldsymbol{\sigma}))}{p_\theta(\sigma \mid \boldsymbol{\sigma})}}_{\text{Unweighted per-symbol contribution}}. \tag{75b}$$

## H. Experimental Setup Details

**Intervention Sampling.** All our interventions count transitions in the corpus; the event function specifies which transitions count. We study two predicates experimentally: **symbol**-level, where the event fires on all transitions emitting a chosen symbol, and **state**-level, where the event fires on all transitions out of a chosen state. These interventions are best described with do-notation introduced in §3. Each of these is captured by some property $\mathcal{P} \subseteq \mathcal{D}$. By intervening on it, we sample according to $\mathcal{D} \sim \mathbb{P}(\cdot \mid \mathrm{do}(\mathcal{P} = p))$, $p \in \mathcal{P}$, where $\mathbb{P}$ denotes the data-distribution probability measure under intervention (formally a Markov kernel; see App. C). For all three interventions, the event function $\phi = (\phi_\lambda, \phi_\delta, \phi_\rho)$ has identically-zero initial and final components ($\phi_\lambda \equiv 0$, $\phi_\rho \equiv 0$); only the transition component varies. Specifically:

*(i)* Symbol intervention on $a_I$: $\phi_{a_I, \delta}(q, \sigma, r) \overset{\text{def}}{=} \mathbb{1}\{\sigma = a_I\}$;

*(ii)* Transition intervention on $\delta_I$: $\phi_{\delta_I, \delta}(q, \sigma, r) \overset{\text{def}}{=} \mathbb{1}\{(q, \sigma, r) = \delta_I\}$;

*(iii)* State intervention on $q_I$: $\phi_{q_I, \delta}(q, \sigma, r) \overset{\text{def}}{=} \mathbb{1}\{q = q_I\}$.

**Ancestral Sampling.** For ancestral sampling, we start the process from the initial state $q_\iota$. We then sample the next symbol by recursively selecting the transitions according to the PFA's probability distribution. For example, given a state $q$ and the transition $q \xrightarrow{\sigma/w} r$, the conditional probability of sampling $\sigma$, i.e. probability $\mathbb{P}(\sigma \mid q)$, is $w$. Sampling terminates at state $q$ with probability $\rho(q)$; otherwise we draw an outgoing transition $q \xrightarrow{\sigma/w} q'$ with probability $w$. The PFA normalization $\rho(q) + \sum_{q \xrightarrow{\sigma/w} q' \in \delta} w = 1$ ensures this is a valid distribution at every state.

**Neural Language Models.** We conduct experiments using both LSTM and transformer-based LMs. The configuration of the neural LMs, including specific hyperparameters, is given in App. I.

**Error bars.** Error bars show one standard error of the mean over the runs grouped at each plotted $x$-coordinate: for causal curves the grouping is by discrete intervention count, and for correlational curves it is by occurrence-count bin (each bin gets its own SEM, independently of the others).

## I. Model and Training Details

We now detail the hyperparameters used for training both the LSTM and transformer models. While they share common settings such as batch size and optimizer, they differ in architectural design and training specifics. The loss we use is over

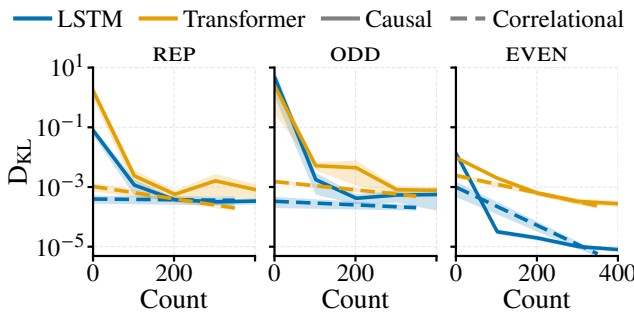

*Figure 7.* Monte Carlo estimates of the state-wise learnability for the automaton given in Fig. 1 using the at-least-once intervention.

the full KL divergence at each symbol position in the training data against the corresponding automaton (formula given in §5); this can be seen as a sample-efficient distillation of the sampled automata.

We use a parameter budget of 128k for both LSTMs and transformers and train all models on CPU. For layer norm, we initialize weights to 1 and biases to 0. When dropout is applicable, we use a value of 0.1. Parameters are uniformly initialized from $[-0.1, 0.1]$. We shuffle data at training, with a max batch size of 128 tokens, and a learning rate of 0.01. We use the Adam optimizer, and clip gradients with a threshold of 5 using $L^2$ norm scaling. The learning rate is multiplied by 0.5 after 5 checkpoints with no decrease in the loss on the validation set. Training is halted after 10 checkpoints if no improvement is observed.

Sampling of an individual dataset typically takes within minutes on a V100 32 GB GPU, as we make use of a GPU for the allsum calculations. The training of an individual dataset typically runs in minutes on a single modern CPU core using the LSTM architecture, while the transformer models can run longer. Our large-scale experiments were run on a cluster with 76 nodes, making use of 16 CPU cores on each node, and occasionally on a single machine with 2 x AMD EPYC 9354 32-Core Processor and two NVIDIA H100. The PARITY + star-free experiment and the varying topology experiment run in under 12 hours, making use of the full CPU/GPU resources for both sampling and training. The larger-scale run took a few days to sample and train in parallel.

## J. Additional Figure

An additional figure for the at-least-once sampling variant of the automaton in Fig. 1 is given in Fig. 7.

