# OpenReview forum: "Causally Evaluating the Learnability of Formal Language Tasks"
_ICML.cc/2026/Conference — ICML 2026 regular_

### Official Review · Reviewer_Z5KU · 2026-03-12

**Soundness:** 3
**Presentation:** 2
**Significance:** 3
**Originality:** 3
**Overall Recommendation:** 4
**Confidence:** 2

**Summary:**

This paper proposes an improved methodological approach for evaluating the learnability of neural language models (e.g., LSTMs and Transformers) on formal languages. The authors argue that conventional correlational evaluations suffer from confounding effects caused by the varying frequencies of other tasks within the training data.
To address this, the paper introduces a novel algebraic structure called the "binning semiring" to define the weights of Weighted Finite-State Automata (WFSAs). This structure provides the significant advantage of allowing efficient tracking and manipulation of probabilities based on the occurrence counts of specific target events, such as reaching a particular state or emitting a specific symbol.
Leveraging the binning semiring, the authors develop a constrained sampling algorithm to generate artificial datasets where the exact number of occurrences of a target event is strictly controlled, effectively implementing a causal intervention (the do-operator). Finally, the models are evaluated by analyzing the relationship between the controlled event occurrences and their true learnability, which is measured by how closely the model's conditional distribution matches the ground truth (using a decomposed KL divergence).

**Compliance With Llm Reviewing Policy:**

Affirmed.

**Final Justification:**

Thank you for the constructive discussion.
After carefully considering this discussion and the authors' final response (second rebuttal), I believe it is appropriate to raise my score to a Weak Accept.

I will outline my reasons below.

First, regarding my concern about the distribution shift: I was initially concerned that the intervention would alter the probabilities depending on the counting, thereby distorting the distribution. However, upon reviewing the paper more closely, I realized that, as stated in Theorem 4.5, the authors have carefully designed the sampling method for string generation so that the distribution is not distorted and perfectly matches the conditioned distribution before the intervention. I admit this might have been an oversight on my part during my initial reading.

Second, regarding the automaton topology: As I pointed out, in probabilistic automata, there can generally exist states that are inherently not strictly distinguishable, even when using the conditional KLD (decomposed KL) proposed by the authors. In such cases, it fundamentally becomes difficult to observe clear differences like those shown in the experimental results, even with interventions.

That said, the authors frankly acknowledged this point in their second rebuttal and promised to explicitly describe it as a limitation. I appreciate this constructive attitude and evaluate it positively.

**Key Questions For Authors:**

- The proposed intervention (using the do-operator to control event counts) inherently alters the training string distribution. How is this distribution shift accounted for when evaluating the model's performance using the proposed Decomposed KL divergence?

**Limitations:**

yes

**Strengths And Weaknesses:**

# Strengths

**Soundness**
The mathematical framework for weighted finite-state automata (WFSAs) using semirings is rigorously defined and systematically presented, which is a commendable aspect of this work. Furthermore, the introduction of the specific "binning semiring" is technically compelling; the authors provide a detailed and sound explanation of how it seamlessly enables the assignment of distinct weights based on the occurrence count of events derived from the automaton's local properties.

**Presentation**
The paper is well-structured and pedagogically effective. By utilizing a concrete toy automaton example (Figure 1), the authors clearly and effectively illustrate why controlling the frequency of specific events (e.g., visits to the Free, Even, and Odd states) is crucial when evaluating the sample complexity required for a model to learn the underlying structure.

**Significance**
The core problem statement—arguing for the necessity of evaluating learnability by strictly controlling the number of state traversals to isolate confounding factors—addresses a highly relevant issue in the field. This perspective is an important contribution to the experimental research community focused on probing the learnability of large language models (LLMs) through the lens of formal languages.

**Originality**
The paper demonstrates clear originality in its methodological approach. Proposing a novel causal framework that leverages WFSAs equipped with the binning semiring to enforce strict event occurrence controls globally across the entire generated dataset is a unique and creative contribution to the study of formal language tasks.

# Weaknesses

**Soundness**
While the mathematical formulation of the semiring framework is sound, there is a fundamental disconnect in the experimental premise. Language models heavily depend on the underlying probability distribution of strings. By strictly controlling the number of event occurrences via the intervention (the do-operator), the proposed method artificially alters the training distribution, causing it to deviate from the original automaton's natural distribution. The paper lacks a rigorous analysis of how this forced distribution shift impacts the intrinsic learning dynamics of the LLM. It remains unclear whether evaluating a model trained on such a distorted distribution against the original ground truth provides a valid measure of its true learnability.

**Presentation**
The limited scope of the experimental results makes it difficult to fully assess the practical effectiveness and scalability of the proposed method. Furthermore, the paper's framing sets aside established theoretical results from the grammatical inference literature, which is difficult to justify. For instance, as demonstrated in foundational work on the PAC learnability of deterministic finite automata (e.g., Alexander Clark), state "distinguishability" is a critical factor in learning automaton structures. The authors assume that states and transitions can be cleanly isolated as independent "tasks" (events). However, there are countless automata where states are highly interdependent and cannot be easily separated in this manner. Omitting this established theoretical context makes the paper's core assumptions less convincing.

**Significance**
The primary focus of the paper is heavily skewed toward the algorithmic mechanics of sampling datasets with controlled event counts. Consequently, the paper falls short of providing novel empirical insights into the actual learning capabilities of LLMs. It does not offer substantial new findings regarding under what specific conditions, or to what extent, modern neural architectures actually learn these formal language events. The contribution remains largely at data generation rather than advancing our understanding of LLM learnability itself.

**Originality**
While the application of the binning semiring to precisely control dataset generation is a novel and creative engineering feat, the originality of the paper is largely confined to this specific sampling technique. Other aspects of the study, including the broader evaluation paradigm and the use of formal languages to probe LLMs, closely follow established conventions in the existing literature.

---

> ### Author Rebuttal · Authors · 2026-03-30
>
> We thank the reviewer for recognizing the rigor of the mathematical framework and the originality of the causal approach. We address each concern below.
>
> ### W1Q: Distribution Shift and Decomposed KL
>
> We want to clarify how the evaluation works, as we believe this directly resolves the concern: **the model is trained on the interventional distribution, but it is evaluated against the original automaton's ground-truth conditionals -- not the interventional distribution.** This is the key design choice, and it is why we developed the decomposed KL divergence (§5, with full derivation in App. E).
>
> Concretely, for a target state $q$, we measure $D_{\mathrm{KL}}(p_{\mathcal{A}}(\cdot \mid q) \| p_{\theta}(\cdot \mid q))$ -- how well the model has learned the next-symbol distribution at $q$ as defined by the original, unmodified automaton. The intervention controls what the model sees during training; the evaluation asks how well it learned each sub-task's true behavior. As stated in §5: "If we were to use the overall KL we would also be considering how other 'tasks' are being learned." The overall KL *would* be distorted by the distribution shift, that is precisely why we do not use it.
>
> One might further ask: does training on a shifted distribution change *how* the model learns the target property? Yes -- and measuring exactly this is the purpose of the causal analysis. We deliberately change how often a property appears in training and then measure how well the model learned it. The resulting relationship between frequency and learnability is the causal effect we seek. In causal inference, an intervention always changes the joint distribution — that is what do(X=k) means. The decomposed KL ensures we evaluate the model against the ground truth regardless.
>
>
> ### W2: Grammatical Inference and State Independence
>
> We appreciate the pointer to the grammatical inference literature. The key distinction is that grammatical inference and our work address **different questions**:
>
> - **Grammatical inference** (Clark et al.): "Given observed strings, can one *recover* the automaton's structure?" Here, state distinguishability is critical because the learner must identify distinct states from data.
> - **Our work:** "Given a *known* ground-truth automaton, does an LM learn its sub-behaviors from data sampled from it?" We never attempt to recover the automaton -- we measure how well the LM approximates known conditional distributions.
>
> Importantly, **we do not assume states are independent.** The model is trained on the full language (all states jointly). The decomposed KL at state $q$ measures: "when the model reaches $q$, does it predict the correct next-symbol distribution?" This is meaningful regardless of state coupling -- just as measuring a student's accuracy on algebra vs. geometry is valid even when mastering algebra helps with geometry. Interdependence between states is not a confound on the measurement; it is part of what we are measuring -- how learning at one state transfers to or interferes with learning at another.
>
> ## W3/W4: Contribution Scope
>
> The empirical results serve a specific purpose: establishing that confounding between formal language tasks is a real phenomenon that systematically distorts learnability estimates. This is the necessary first step — demonstrating the problem exists and that our tools isolate it. Whether any particular prior finding is affected depends on the specific automaton and property studied; our tools exist precisely to check that. Prior to this work there was no principled way to distinguish a genuine learnability result from a confounding artifact in formal language evaluation. What researchers do with it — including revisiting prior findings — is the next step, and deliberately outside the scope of this paper.
>
> Regarding empirical scope: Setting 2 samples 1,000 random PFSAs with 50 states and 10 symbols — this is among the larger formal language evaluation studies we are aware of. The confounding mechanism is not specific to regular languages; extension to context-free languages is a natural next step.
>
> Regarding originality: The binning semiring is not an application of an existing tool — it is an algebraic structure invented to make the analysis tractable, with a non-trivial Kleene closure derivation (Proposition C.4, Lemma C.5). The constrained sampling theorems (4.3–4.5, D.1–D.3) are new results, as is the decomposed KL as a targeted evaluation metric. The causal framework is established — but the framework is the scaffolding, not the contribution. Everything required to instantiate it in this setting is new.

---

> > ### Author Rebuttal · Reviewer_Z5KU · 2026-04-03
> >
> > Thank you for your detailed rebuttal. I appreciate the explanation and fully acknowledge the theoretical and algorithmic contributions of your framework.
> >
> > The rebuttal confirms that my initial understanding of the intervention mechanism and the proposed Decomposed KL formulation (W1) was accurate. Regarding W2, I am fully aware that your work does not attempt to solve the grammatical inference problem (i.e., recovering automaton structure from data), but rather uses known automata for sampling.
> >
> > With these premises explicitly confirmed, I would like to reiterate and clarify the true intent of my critiques:
> >
> > **(W1) The Issue of Evaluating on a Shifted Distribution**
> > While I understand the intent behind evaluating against the ground-truth conditional distribution to isolate tasks, when an artificial constraint—such as appearing "exactly $N$ times"—is imposed on the dataset, autoregressive models (LSTMs/Transformers) learn to inherently warp their *local* transition probabilities to satisfy this *global* constraint. Therefore, a model that perfectly fits the intervened training data will naturally diverge from the original, unconstrained local conditionals.
> >
> > Consequently, even when measuring the next-token prediction at the target state (Decomposed KL), the model has already distorted its local predictive probabilities to meet the intervention constraint (exactly $N$ times), meaning that comparing it to the original probabilities will inevitably result in a divergence (penalty). The authors argue that the gap observed in Setting 2 demonstrates the elimination of confounding. However, the presented empirical results fail to rule out the possibility that this gap merely reflects the penalty (error) caused by the local predictive probabilities being warped by the global intervention constraint.
> >
> > **(W2) The Generality of the Empirical Validation**
> > My reference to "distinguishability" was not about structural inference, but about the generality of the experimental setup. The empirical demonstrations (e.g., Figure 1) use highly simple automata where states are cleanly separable. However, in general automata, states are usually highly entangled and inherently difficult to distinguish.
> >
> > I certainly understand that this work represents an important and promising first step and raises a very valuable point. However, I believe the paper would be significantly stronger if it included a solid discussion or theoretical analysis clarifying exactly under what types of automata the proposed method remains effective.
> >
> > Thank you again for your time and the engaging discussion.

---

> > > ### Author Response · Authors · 2026-04-03
> > >
> > > We sincerely thank the reviewer for the constructive follow-up and for confirming that the original concerns are resolved. We address the two clarified points below.
> > >
> > > **W1: On the interpretation of the Decomposed KL under intervention**
> > >
> > > The specific concern — that the gap between causal and observational curves might reflect a "penalty" from training on a constrained distribution rather than the removal of confounding — can be ruled out structurally, as we show below. Separately, we agree that the paper should be more explicit about what the total causal effect captures and will add a discussion of this.
> > >
> > > Since the model has shared parameters, performance at a target state reflects the entire training distribution, including indirect effects from the frequencies of other states. This is precisely why the causal analysis is needed: the joint distribution over state frequencies is determined by the automaton's topology and weights, so automata where the target count is naturally $n$ will systematically differ in their frequencies at other states as well. Under observation, these automata are overrepresented at $\text{count}=n$, conflating the effect of target frequency with these correlated structural properties. The causal curve removes this confounding — and the gap between the two curves can be shown to arise entirely from this reweighting, not from any penalty introduced by the intervention.
> > >
> > > The key observation is that for a fixed automaton $A$, the distribution over datasets $P(D \mid A, \text{count}=n)$ is identical whether $\text{count}=n$ arose naturally or was enforced by the constrained sampler. The sampler does not change how data is generated from a given automaton at a given count — it only changes which automata contribute to the estimate. Defining $M_q(A, n) = \mathbb{E}\_{D \sim P(D \mid A, \text{count}=n)}[ \mathrm{DKL}(p_A(\cdot \mid q) | p_\theta(\cdot \mid q)) ]$, the two estimands are:
> > >
> > > $\mu_{\mathrm{int}}(n) = \mathbb{E}\_{A \sim P(A)}[ M_q(A, n) ], \quad \mu_{\mathrm{obs}}(n) = \mathbb{E}\_{A \sim P(A \mid \text{count}=n)}[ M_q(A, n) ].$
> > >
> > > Because $M_q(A, n)$ is shared, any effect of the count constraint on the model's learning, including the distributional warping the reviewer describes, is already inside $M_q(A, n)$ and cancels in the comparison. The gap reduces to:
> > >
> > > $\mu_{\mathrm{int}}(n) - \mu_{\mathrm{obs}}(n) = \int M_q(A, n) \left[ P(A) - P(A \mid \text{count}=n) \right] dA,$
> > >
> > > which is purely a reweighting over automata, not an artifact of the intervention mechanism.
> > >
> > > This is consistent with the empirical pattern in Setting 2: at low occurrence counts, the observational and causal curves diverge and the observational data exhibits an inverse trend, reflecting precisely the confounding described above — automata where the target event is naturally rare are structurally different from typical automata forced to the same count.
> > >
> > > We will add a discussion clarifying what the total causal effect captures — the expected effect of target frequency on target performance, averaging over the natural distribution of automata — and what it does not isolate without further analysis, namely the direct effect of target frequency versus indirect effects mediated through other states.
> > >
> > > **W2: On automaton topology**
> > >
> > > We agree that a discussion of how automaton structure affects interpretability would strengthen the paper, and we will add one.
> > >
> > > We want to note that the method itself — the constrained sampler, causal framework, and decomposed KL — is valid for any DFSA; there is no topology where it produces incorrect results. Settings 2 and 3 already use general, densely connected automata (1,000 random 50-state PFSAs and a single 40-state automaton with 400 weight configurations), and the causal–observational discrepancy persists throughout.
> > >
> > >
> > > That said, automaton structure does affect what the total causal effect means. When the target state is reachable via many paths through tightly connected components, changing its visit frequency will tend to co-vary with frequencies at neighboring states, so the total causal effect will include substantial indirect contributions. When the target state is more structurally isolated, the total effect will be dominated by the direct effect of target frequency. We will discuss these conditions explicitly and note that mediation analysis (controlling for frequencies at non-target states) is a natural extension for disentangling direct from indirect contributions. We will also discuss how properties such as the number of paths reaching the target state, the connectivity of the topology, and the probability of the target event under the natural distribution affect the degree of confounding.
> > >
> > > ---
> > >
> > >
> > > We hope that the structural argument resolving W1 and the promised discussion for W2 address the reviewer's remaining concerns. We believe these additions strengthen the paper, and we would be grateful if the reviewer would consider re-evaluating their score.

---

### Official Review · Reviewer_zuLg · 2026-03-13

**Soundness:** 3
**Presentation:** 3
**Significance:** 3
**Originality:** 3
**Overall Recommendation:** 5
**Confidence:** 3

**Summary:**

The paper proposes a causal framework for evaluating the learnability of formal language tasks by neural language models. It argues that existing evaluations are correlational because task frequency in training data can be confounded by other properties of the generating process. To address this, the authors introduce a binning semiring and constrained sampling algorithm enabling causal interventions on training data. Experiments with LSTM and Transformer language models on probabilistic finite-state automata show that causal estimates of task learnability can differ substantially from observational ones, demonstrating the importance of causal evaluation.

**Compliance With Llm Reviewing Policy:**

Affirmed.

**Final Justification:**

Thank the authors for addressing my concerns and for the clarifications provided. I will maintain my positive score.

**Key Questions For Authors:**

Could alternative causal interventions provide similar insights?

**Limitations:**

Yes

**Strengths And Weaknesses:**

Strengths:

1. The paper introduces a principled causal perspective for evaluating the learnability of formal language tasks.

2. The proposed binning semiring and constrained sampling algorithm provide an elegant and theoretically grounded mechanism.

Weaknesses:

1. The study focuses on formal languages generated by probabilistic finite-state automata. While this setting enables controlled causal analysis, it remains unclear whether the conclusions about causal learnability and task frequency transfer to natural language settings.

2. It remains unclear how the proposed methodology provides practical values.

---

> ### Author Rebuttal · Authors · 2026-03-30
>
> We thank the reviewer for the kind words and positive review.
> ### W1: Natural Language Transfer
> The core insight -- that task frequency is confounded with other properties of the data-generating process -- applies to any multi-task learning setting, including natural language. For example, a corpus with many relative clauses will also differ from one with few in sentence length, vocabulary, and the frequency of other syntactic constructions, making it hard to attribute differences in learnability to any single property. Our paper establishes a necessary first step: demonstrating that causal and correlational analyses can differ substantially, using a setting where we have ground truth and can verify this rigorously. Extending to NL is future work – and it would require approximate methods. But before developing approximate NL tools, one needs to establish that the problem is real, which is what we do.
>
>
> The formal language analysis also has standalone value: it enables fair comparison of language model architectures. If a correlational study finds that architecture A outperforms B on a task, our work shows this conclusion may be confounded. Causal evaluation resolves this, allowing reliable architecture comparisons even without involving natural language. More generally, formal languages idealize the kind of multi-task structure found in NL, making them a natural controlled setting for studying confounding effects that are likely present (but much harder to isolate) in natural data.
>
> ###  W2: Practical Value
> For researchers conducting formal language evaluations of neural LMs (an established line of work) our framework provides a complete pipeline for avoiding confounded conclusions. The specific practical finding is that observational estimates can produce non-monotonic learnability curves that are *confounding artifacts*, not genuine phenomena (Figures 2, 5, 6). In the large-scale setting (Figure 5), the discrepancy is particularly striking: at low occurrence counts, the observational trend *inverts*. This happens because automata that naturally produce few instances of a property tend to do so *because* of their structure -- and that same structure affects learnability in other ways, creating a confound between scarcity and difficulty. Any study correlating task frequency with model performance should verify such patterns under causal evaluation. This principle applies whenever training data composition correlates with auxiliary properties, a condition that can be assumed to hold for much of natural data.
>
> ### Q: Could alternative causal interventions provide similar insights?
> Yes, and the paper already demonstrates this. We implement two intervention types: exact-count (the property occurs exactly $k$ times) and at-least-once (the property appears in at least $k$ strings). These answer different questions: exact-count provides fine-grained dose-response curves for task frequency, while at-least-once asks whether mere per instance exposure to a property matters regardless of frequency. Both show causal-observational divergence (main text and Fig. 7).
>
> More broadly, the causal graphical model (sec. 3) is defined independently of how the intervention is implemented. The event function (Definition 4.1) can target arbitrary sets of transitions -- all transitions emitting a given symbol, all transitions entering a set of states, or all scanning transitions (which corresponds to string length). Beyond what the current binning semiring supports, one could also design interventions on composite properties (e.g., "visit state A and then state B") or joint interventions controlling multiple properties simultaneously. These would require new sampling machinery (an avenue for future work), but the causal framework and decomposed KL evaluation would remain unchanged. We view this separation of the causal framework from the sampling mechanism as a strength of the design.

---

> > ### Author Rebuttal · Reviewer_zuLg · 2026-04-04
> >
> > I thank the authors for addressing my concerns, and I will maintain my positive score.

---

### Official Review · Reviewer_4168 · 2026-03-15

**Soundness:** 3
**Presentation:** 3
**Significance:** 2
**Originality:** 2
**Overall Recommendation:** 3
**Confidence:** 3

**Summary:**

The paper addresses a fundamental flaw in how the learnability of neural language models is evaluated: the reliance on correlational data that fails to account for confounders between tasks. The authors propose a causal graphical model for learnability and introduce the binning semiring, an algebraic tool that enables targeted sampling of datasets with exact constraints on task occurrences while keeping other language properties fixed. Through three case studies, they demonstrate that "causal" sample complexity—the true amount of data needed to learn a task—can differ significantly from "observational" estimates.

**Compliance With Llm Reviewing Policy:**

Affirmed.

**Key Questions For Authors:**

Your framework relies on exactly countable properties in PFSA-generated languages. How would the method extend to natural language tasks where properties are noisy, ambiguous, or not perfectly observable (e.g., reasoning, syntax, semantics)? What modifications would be required?

**Limitations:**

The paper does not include a dedicated "Limitations" section, but the authors self-identify several constraints and technical boundaries throughout the text.

**Strengths And Weaknesses:**

Significance
Strength: Identifies a fundamental flaw in how we measure "learnability." By proving that observational data can lie about how much data a model needs, it sets a new standard for rigor in interpretability research.

Weakness:
- Scalability Concerns. the framework relies on properties being exactly countable/algebraic. It is currently unclear how this translates to natural language where "tasks" (e.g., sentiment, syntax, reasoning) are latent and highly entangled.
- Impact on Practical Development. While the paper demonstrates why current evaluation fails, it does not yet show how a developer would use these causal insights to build better models, potentially limiting its immediate utility to practitioners.

Originality
Strength: The introduction of the binning semiring is a distinct algebraic contribution that provides a mathematically grounded way to perform interventions in formal languages.

Weakness: The work is a sophisticated application of established causal inference tools (graphs and interventions) to a new domain (formal languages), rather than an invention of new causal primitives.

Soundness
Strength: The sampling algorithms are proven correct, and the experiments across 1,000 machines provide high statistical confidence in the formal language results.

Weakness: Domain Narrowness. The empirical study is restricted to toy automata. While this is the standard for "science of DL" papers, it leaves open the question of whether these causal "interference" effects remain the dominant factor in more complex, non-regular languages.

---

> ### Author Rebuttal · Authors · 2026-03-30
>
> We thank the reviewer for the thoughtful assessment.
>
> ### W1 and W4: Scalability and Extension to Natural Language
> The core insight — that task frequency is confounded with other properties of the data-generating process — applies to any multi-task learning setting, including natural language. However, doing this kind of causal analysis on a natural language corpus would be very difficult. Consider studying how the frequency of relative clauses affects an LM's ability to learn them. To intervene causally, you would need to construct corpora that vary relative clause frequency while holding everything else constant. The number of potential confounders is vast — sentence length, vocabulary, style, the frequency of other syntactic constructions — and each one would need to be identified and controlled for. Even then, you would have no ground-truth model to verify that your intervention actually isolated the target property. Existing causal approaches in NL (e.g. Vig et al., 2020; Finlayson et al., 2021, Chen et al., 2024) don’t provide principled control over corpus-level confounding across the full data-generating distribution.
>
> PFSAs give us a setting where this analysis is possible, using the same neural LM architectures used for NL. This is an insight that can only be rigorously established where ground truth is available, and formal languages provide exactly that. A long line of work uses them precisely this way (Lake & Baroni, 2018; Deletang et al., 2023; Valvoda et al., 2022; Borenstein et al., 2024, inter alia).
>
> We see three modifications that could enable NL application: (1) **Approximate property labels** via syntactic parsers or POS taggers, where label noise preserves the causal logic provided it is not itself confounded with the target property.
> (2) **Importance-weighted subsampling** of a large corpus by approximate property frequency, paralleling propensity score weighting.
> (3) **Probing-based evaluation** in place of decomposed KL, using held-out data or probing classifiers instead of a known automaton.
>
> Finally, the confounding mechanism is not specific to regular languages — extension to context-free languages is a natural next step.
>
> ### W2: Practical Utility
> Formal languages are used precisely because they yield deep, transferable insights about neural architectures — for instance that Transformers struggle with the parity language while LSTMs do not (Hahn & Rofin, 2024), or that neither architecture handles the full Chomsky hierarchy uniformly (Deletang et al., 2023). These findings directly inform our understanding of what modern LMs can and cannot do.
>
> Our contribution is that such findings may be confounded. A conclusion like "architecture A learns property X better than architecture B" drawn from correlational data could reflect the fact that A's training distribution happened to co-vary with a helpful structural feature, not a genuine architectural advantage. A researcher observing any such pattern — including non-monotonic learnability curves, architecture-specific failure modes, or sample complexity differences — now has a tool to determine whether it is a genuine architectural property or a confounding artifact. Our results (Figures 2, 5, 6) show the latter is a real risk. Causal evaluation is therefore not an abstract methodological nicety; it is the prerequisite for trusting the architectural conclusions that this entire line of research is designed to produce.
>
> ### W3: Originality
> We want to respectfully push back on the characterization of our originality as "applying established causal inference tools to a new domain." The binning semiring is not an application of an existing tool; it is an algebraic structure invented to make the causal analysis tractable. Representing transition weights as polynomials over a cyclic monoid, with multiplication defined as a truncated Cauchy convolution, is not a standard construction. The Kleene closure in this semiring requires a new closed-form derivation. Proposition C.4 and Lemma C.5 establish existence and uniqueness of this solution. These are not routine. The constrained sampling theorems (4.3–4.5 and D.1–D.3) are new results. Theorem 4.4 in particular — deriving the conditional distribution over per-string event counts given a whole-dataset constraint — requires a combinatorial argument specific to the binning semiring structure. No prior work gives this. The KL decompositions — evaluated at the state, transition, and symbol level — is new. Without them, measuring the causal effect of an intervention would be confounded by how other tasks are simultaneously being learned.
>
> We agree that the causal framework (DAG, do-operator, adjustment formula) is established. But the framework is the scaffolding, not the contribution. The contribution is everything required to instantiate it in this setting, and none of that existed before this work.

---

> > ### Author Rebuttal · Reviewer_4168 · 2026-04-05
> >
> > Thanks very much for the detailed responses! I feel the key limitation of this paper -- its application scope of formal languages generated by probabilistic finite-state automata and lack of clear transferability to natural language tasks, is not addressed. I will keep the current score.

---

> > > ### Author Response · Authors · 2026-04-08
> > >
> > > Artificial tasks are widely used for studying language models. As we pointed out in the rebuttal above, formal languages are the standard for studying learnability (Deletang et al., 2023; Hahn & Rofin, 2024; Strobl et al., 2024). Artificial tasks are also used in other subfields, such as mechanistic interpretability (Nanda et al., 2023; Li et al., 2023; Brinkmann et al., 2024), where no one questions the absence of natural language validation. Even general reasoning tests such as ARC-AGI (Chollet 2019, 2026) use synthetic data. Requiring natural language validation thus goes against the grain of more than one subfield of language model research.
> > >
> > >
> > >
> > > To recap, our work demonstrates that the standard approach for evaluating language model architecture learnability is flawed and introduces a significant technical and methodological contribution to address it. We do so within the formal language framework because that is what the field is already doing, and for good reasons. Full control is what enables laboratory-grade study of language model architectures.
> > >
> > >
> > > All three other reviewers engaged with these arguments, marked their concerns as fully resolved, and did not consider natural language transferability grounds for rejection. We hope that this clarification, combined with the responses in the prior rebuttal, has addressed each of your concerns.

---

### Official Review · Reviewer_QhHj · 2026-03-17

**Soundness:** 3
**Presentation:** 2
**Significance:** 2
**Originality:** 3
**Overall Recommendation:** 4
**Confidence:** 3

**Summary:**

This paper argues that learnability studies on formal languages should be evaluated causally rather than purely correlationally. The key point is that, when training language models on data generated from probabilistic finite-state automata (PFSAs), the frequency of a target property can be confounded by other structural properties of the automaton and dataset. To address this, the paper introduces a graphical causal model for the pipeline from ground-truth automaton to sampled dataset, trained model, and performance. It then develops a technically nontrivial constrained sampling framework based on a binning semiring, which allows datasets to be sampled under exact or at-least event-count constraints while keeping other aspects of the generating process fixed. The paper also derives decomposed KL objectives that target the learnability of specific states, transitions, or symbols. Experiments on several PFSA settings with LSTMs and Transformers show that causal estimates of learnability can differ substantially from observational ones, suggesting that standard correlational analyses may mischaracterize sample complexity and task difficulty.

**Compliance With Llm Reviewing Policy:**

Affirmed.

**Final Justification:**

The authors' rebuttal fully resolved my concern. Considering the questions and rebuttals from other reviewers, I'm keeping my score toward acceptance.

**Key Questions For Authors:**

1. **How much of the empirical gain from the proposed framework depends on the full binning-semiring machinery, as opposed to simpler constrained-sampling baselines?**
   A comparison to simpler alternatives would help clarify whether the main contribution is primarily conceptual, computational, or both. If strong empirical or runtime advantages over simpler baselines are shown, my confidence in the practical significance would increase.

2. **How sensitive are the conclusions to the precise definition of the intervened property?**
   The current framework targets countable events over transitions, states, or symbols. It would help to know whether similar causal/observational discrepancies persist for more composite or nonlocal properties. A positive answer would make the contribution feel broader and more compelling.

3. **What exactly is meant by a “formal language task” throughout the paper?**
   The introduction suggests that tasks are defined structurally as parts of the automaton, such as connected subgraphs or locations in the automaton, and Figure 1 informally discusses the substring $aa^*$ as a task. However, later sections seem to operationalize learnability more locally in terms of states, transitions, symbols, or sets of transitions. It would help to define whether a “task” in the paper should be understood as an entire sublanguage, a local automaton component, or a more general property defined over transitions at the beginning of the paper.

**Limitations:**

Yes

**Strengths And Weaknesses:**

### Strengths

- **Soundness:** The paper is built around a clear and well-motivated question: whether observed task frequency truly reflects task learnability in formal-language settings. The causal framing is principled, and the technical development is substantial. In particular, the graphical model, the constrained sampling procedure, and the decomposed KL metric are aligned with the stated goal of isolating task-specific learnability. The empirical section also evaluates the method in multiple settings rather than relying on a single toy example.

- **Presentation:** The paper has a coherent high-level arc from motivation, to causal formulation, to sampling machinery, to empirical case studies. The examples in the figures help communicate the main qualitative takeaway that causal and observational curves can differ materially.

- **Significance:** The work identifies a genuine methodological issue in a line of research that often uses formal languages as controlled testbeds for understanding neural language models. If the paper’s argument is correct, then some past conclusions about what architectures can or cannot learn from synthetic formal-language datasets may depend on uncontrolled confounding. That makes the contribution meaningful for researchers using such benchmarks.

- **Originality:** The combination of a causal intervention view, an algebraic constrained-sampling mechanism via the binning semiring, and targeted decomposed KL evaluation feels novel. Even if each ingredient has intellectual precedents, the overall formulation is distinctive and well tailored to the problem.

### Weaknesses

- **Soundness:** While the technical machinery is interesting, it is not always clear which parts are essential for the main empirical conclusion and which parts are primarily mathematical generalization. The paper would be stronger with more direct comparisons against simpler alternatives, such as less general constrained sampling procedures or carefully controlled rejection-sampling baselines, to clarify what the binning-semiring machinery buys in practice beyond asymptotic discussion. Relatedly, the empirical results mainly show qualitative separation between observational and causal curves, but there is less analysis of sensitivity to training hyperparameters, dataset size, model size, or automaton size.

- **Presentation:** The paper is fairly dense. The notation load is high, and the transition from the causal framing to the semiring construction to the sampling theorems is demanding. I found the overall story understandable at a high level, but the technical path is harder to follow than ideal, especially for readers who are not already comfortable with weighted automata and semiring-based reasoning. Some readers may also find the appendix necessary rather than supplementary.

- **Significance:** The contribution is important within the formal-language evaluation literature, but its broader impact is still somewhat limited at this stage. The paper itself acknowledges that extending the framework to natural language would require approximate labels for target phenomena, and the experiments are explicitly illustrative rather than a comprehensive empirical study of architectures. As a result, the paper currently reads more as a strong methodological contribution for a specialized setting than as something that immediately changes broader LM evaluation practice.

---

> ### Author Rebuttal · Authors · 2026-03-30
>
> We thank the reviewer for the constructive evaluation.
> ### Q1: Binning Semiring Necessity vs. Simpler Baselines
>
> The binning semiring and the sampling algorithm we develop are essential for running our experiments. We considered rejection sampling and an approach that intersects the PFSA with a count-tracking automaton for each target count, then run weight pushing, this works but is expensive ($\mathcal{O}(|\mathcal{Q}|^3 N^4)$ as analyzed in the paper). Rejection sampling requires generating an entire dataset, checking whether the target event count matches (which is extremely unlikely), and starting over if it does not. The binning semiring solves this by sampling directly from $P(\text{dataset} \mid \text{count} = k)$, computing exact conditional probabilities via the lifted automaton's backward weights.
>
> **Concrete comparison:** Section 4 in the main paper reports an asymptotic improvement from $\mathcal{O}(|\mathcal{Q}|^3 N^4)$ to $\mathcal{O}(\max(K, |\mathcal{Q}|^3) N^2)$. In additional benchmarks on 100-state, 10-symbol PFSAs generating 1,000 strings, rejection sampling averaged ~10 min (worst case 22 min); our method averaged ~20 sec (worst case 25 sec), we will add this to the paper. For state-visit count constraints across entire datasets, rejection sampling is infeasible.
>
> Regarding prior art. The closest algorithm that we're aware of is the length-constrained sampling algorithm of MLRegTest (Van der Poel et al. 2024), which handles only string *length* constraints, only individual strings (rather than constraints across the whole dataset), only samples from the uniform distribution, and does not handle arbitrary PFSA distributions. We can mention this in the paper.
>
> The contribution is both **conceptual** (the causal framework and decomposed KL) and **computational** (the binning semiring makes it tractable).
>
> ### Q2: Sensitivity to Property Definition
>
> The experimental settings demonstrate robustness across different kinds of PFSA properties:
>
> - **Setting 1 (Figure 2):** Intervenes on transitions entering the parity vs. free branch -- effectively a sublanguage-level property.
> - **Setting 2 (Figure 5):** Intervenes on *symbol-level* occurrences and *state-level* visits across 1,000 diverse PFSAs.
> - **Setting 3 (Figure 6):** Intervenes on *state-level* visits in a single 40-state automaton across 400 weight configurations.
>
> Causal-observational discrepancies persist across all settings, including Setting 1 which already constitutes a multi-transition composite property. More broadly, we argue that the confounding mechanism is agnostic to property complexity. Confounding arises because any property whose frequency varies across automata will co-vary with other structural features – both are determined by the same topology and weights. This holds regardless of whether the property targets a single transition or a composite pattern.
>
>
> ### Q3: "Formal Language Task" Definition
>
> A "formal language task" is any property of strings definable as a function of the transitions traversed during generation, formalized by the event function in Definition 4.1:
>
> - **Symbol-level:** "how often does symbol $a$ appear?"
> - **Transition-level:** "how often does transition $q \xrightarrow{a} q'$ fire?"
> - **State-level:** "how often is state $q$ reached?"
> - **Sublanguage-level:** "does the string enter the parity branch?" (Setting 1)
> - **Composite:** Any set of transitions can be targeted via an event function
>
> We agree the paper creates some ambiguity between the informal notion of "task" in the introduction and the formal operationalization. The connection is that the informal notions are special cases of targeting sets of transitions. We will add a clear definition linking the two in the revised version.
>
> ## Additional Concerns
>
> **Sensitivity to parameters:** The difference between causal and correlational, which is the key point is consistent across automaton topologies, weight configurations, and two architectures (LSTM, Transformer), suggesting the phenomenon is general rather than an artifact of specific training choices. Since the gap between causal and observational persists, we expect it to be robust to model size or hyperparameters. Dataset size is fixed by design since varying it alongside task frequency would reintroduce the confounding we aim to isolate, since task frequency is jointly determined by the automaton's parameters and dataset size.
>
> **Presentation:** We take this concern seriously. The paper does include a concrete worked example (E.g. Figure 4) that threads through from the PFSA to the binning automaton with actual numeric weights. For the camera-ready, we will add bridging intuition before the formal semiring definitions, connecting each algebraic step back to the sampling goal, so that the transition from causal framing to semiring construction is smoother for readers less familiar with weighted automata. We will also make the connection to the appendix stronger.

---

> > ### Author Rebuttal · Reviewer_QhHj · 2026-04-01
> >
> > The rebuttal strengthens the case for the necessity of the binning-semiring machinery by clarifying both the computational motivation and the limitations of simpler baselines. It also clarifies the intended definition of a “formal language task,” which helps resolve an ambiguity in the original draft. I still find the evidence for robustness across broader classes of properties and training settings somewhat incomplete: the authors only provided a reasonable argument and point to diversity across their existing settings. The presentation concern is acknowledged constructively, though it remains a weakness of the current version.

---

> > > ### Author Response · Authors · 2026-04-03
> > >
> > > We sincerely thank the reviewer for the careful engagement and for confirming that the original concerns are resolved.
> > >
> > >
> > > On robustness: we refer to our argument in the rebuttal that the confounding mechanism is structural and agnostic to property complexity — it arises whenever a property's frequency is jointly determined by the automaton's topology and weights, which guarantees co-variation with other state frequencies. Extending to broader property classes is a natural direction for future work, and one the framework directly supports.
> > >
> > >
> > > On presentation: we take this seriously and will follow through on the promised revisions to bridge intuition before the formal semiring definitions and smoother connections to the appendix in the updated version.

---

### Decision · Program_Chairs · 2026-04-30

**Decision:**

Accept (regular)

**Comment:**

This paper has sparked extended and deep discussion in the reviews and rebuttals. This is always a sign that there is an interesting contribution here. The Paper studies LLMs only applied to a synthetic setting of Formal languages; this line of research has been pursued before but what is novel here is the study of what happens in multi-task settings and an analysis done through a causal lens. There are 2 classes of objections raised:

1. methodological: The discussion here was extensive, and imo we are left with no major concerns. In the final round of disucssions, the following points were made in order to address lingering concerns (shared here FTR and so that authors can take it into account):

"for a fixed automaton and fixed target count/property, the constrained sampler matches the same conditional dataset distribution as natural sampling conditioned on that event. If so, then the gap between the observational and interventional curves is not explained by an additional artifact of the intervention itself. Rather, the remaining systematic difference is that the two quantities average over automata differently: observation reweights toward automata that naturally realize the target count, while intervention preserves the original automaton distribution.

I do think the exchange reveals an important nuance that should be stated more clearly in the paper: the estimated quantity is best understood as a total causal effect of enforcing a target property on target performance, rather than a fully isolated direct effect of target frequency alone. I view that as a limitation on interpretation rather than a fatal soundness issue. "

"..as stated in Theorem 4.5, the authors have carefully designed the sampling method for string generation so that the distribution is not distorted and perfectly matches the conditioned distribution before the intervention"

However, there is a minor limitation that authors acknowledged : "in probabilistic automata, there can generally exist states that are inherently not strictly distinguishable, even when using the conditional KLD (decomposed KL) proposed by the authors. In such cases, it fundamentally becomes difficult to observe clear differences like those shown in the experimental results, even with interventions."

2. This work applies established causal reasoning tools to LLMs rather than proposes new ones. As such, it could be seen as a "straightforward" apply technique A from field B to problem Z paper, which *slightly* diminishes the novelty value.

3. Finally, the question of importance of the paper, given no "real" tasks were addressed. This is a bit of a judgement call, but in my view, in a world where so much real cutting edge application work happens in unpublished frontier lab settings, the role of academic research should shift even more towards this type of research i.e. *understanding* our models better even if there is no immediately clear application of the knowledge.